# Psychophysical reverse correlation reflects both sensory and decision-making processes

Gouki Okazawa [1], Long Sha[1], Braden A. Purcell[1] & Roozbeh Kiani [1,2,3]

Goal-directed behavior depends on both sensory mechanisms that gather information from the outside world and decision-making mechanisms that select appropriate behavior based on that sensory information. Psychophysical reverse correlation is commonly used to quantify how fluctuations of sensory stimuli influence behavior and is generally believed to uncover the spatiotemporal weighting functions of sensory processes. Here we show that reverse correlations also reflect decision-making processes and can deviate significantly from the true sensory filters. Specifically, changes of decision bound and mechanisms of evidence integration systematically alter psychophysical reverse correlations. Similarly, trial-to-trial variability of sensory and motor delays and decision times causes systematic distortions in psychophysical kernels that should not be attributed to sensory mechanisms. We show that ignoring details of the decision-making process results in misinterpretation of reverse correlations, but proper use of these details turns reverse correlation into a powerful method for studying both sensory and decision-making mechanisms.

[1] Center for Neural Science, New York University, New York, NY 10003, USA. [2] Department of Psychology, New York University, New York, NY 10003, USA. [3] Neuroscience Institute, NYU Langone Medical Center, New York, NY 10016, USA. Correspondence and requests for materials should be addressed to R.K. (email: roozbeh@nyu.edu)

Accurate characterization of behavior is key to understanding neural computations[1,2]. Not only do we want to know which behaviors arise from sensory inputs in an environment, but also we need to understand the mechanisms through which sensory inputs lead to behavioral outputs. Over the past few decades, several system identification techniques have been developed to address these needs. Among the most commonly used techniques is psychophysical reverse correlation[3–5], a technique that aims to estimate how sensory information is weighted to guide decisions. The core idea is that by quantifying the stimulus fluctuations that precede each choice (i.e., reverse correlation), one can infer the spatiotemporal filter implemented by the sensory processes (Fig. 1). It can be shown mathematically that under the assumptions of signal detection theory (SDT) for the decision-making process, psychophysical reverse correlation does recover the true sensory weights[6,7]. In SDT, a linear filter is applied to a sensory stimulus and the outcome is compared to a decision criterion. The result of this comparison (higher or lower than the criterion) dictates the choice[8]. If stimuli on different trials are drawn from a symmetric distribution (e.g., Gaussian), reverse correlation will accurately estimate the linear sensory filter of SDT by averaging the stimuli that precede a particular choice.

The technique can also be extended to the temporal domain to recover the dynamics of the weighting function when choices are based on filtering a sequence of observations and comparing the results to a criterion[4,9]. These temporal extensions resemble spike-triggered averaging techniques, which derive spatio-temporal receptive fields (linear kernels) of spiking neurons[10–12], under the assumption that firing rates are determined by filtering sensory inputs followed by application of a static nonlinearity. In general, when a discrete outcome arises from a sequence of linear and nonlinear computations, reverse correlation is a recommended method for estimating the linear component of the computation. How well does this recommendation work in practice for sensory decisions?

Studies of the decision-making process over the past decade have revealed that the simple assumptions of SDT do not adequately capture the complexity of perceptual decisions. We now know that for many decisions, subjects integrate sensory evidence in favor of different choices, and the final decision is made when the integrated evidence reaches a satisfactory threshold[13–16]. Several key features of this process are absent in simple temporal extensions of SDT. First, subjects can flexibly adjust their decision bound within and across trials to change how much evidence to integrate, and thereby trade off the accuracy and speed of their decisions[17,18]. Second, neural implementation of the decision-making process relies on a competition or race between multiple integrators, rather than reaching a decision bound in a single integrator. Third, realistic implementations of these computations in neural networks require taking into account biophysical constraints (e.g., lower limit of firing rates at zero[19–21]) and network mechanisms of integration (e.g., mutual inhibition[19,22,23]). Finally, applying theory to real experimental data requires taking practical limitations into account. A key factor that has been largely ignored thus far is the sensory and motor delays (non-decision time). The sum of the non-decision time and the time spent on integration of evidence (decision time) determine experimentally measured reaction times (RTs)[24,25]. Because the non-decision time limits the relevant stimulus history for the choice, it could distort the outcome of reverse correlation. How much do these factors influence the estimation of sensory filters with psychophysical reverse correlation? Except for scant examples in the past literature that studied basic properties of the integration of sensory evidence (e.g., bounded or leaky accumulation)[19,26–28], the answer is largely unknown. A systematic exploration is timely because mechanistic studies of sensory and decision-making processes have become a cornerstone of modern neuroscience and because accurate methods for quantifying the relationship between experimental stimuli and behavior provide a critical foundation for these investigations[2].

We show that psychophysical reverse correlation deviates qualitatively and quantitatively from sensory weights under several variants of decision-making models. Experiments in which stimulus-viewing duration is controlled by the experimenter often do not allow distinguishing these variants, leaving the mechanistic cause of observed kernel dynamics obscure, unless special measures are implemented (e.g., variation of stimulus durations across trials). RT tasks, where the stimulus-viewing duration is controlled by the subject and reaction times can be directly measured by experimenters, offer much more leverage, especially when a model-based approach is adopted to correct for expected deviations of the reverse correlation from sensory weights. We show that these deviations are not caused by the presence of a decision bound. Rather, they emerge from the presence of variable sensory and motor delays, changes of decision bound within and across trials, lower limits for accumulated evidence, integration time constants, and mutual inhibition of competing accumulators. Knowing about these deviations enables us to

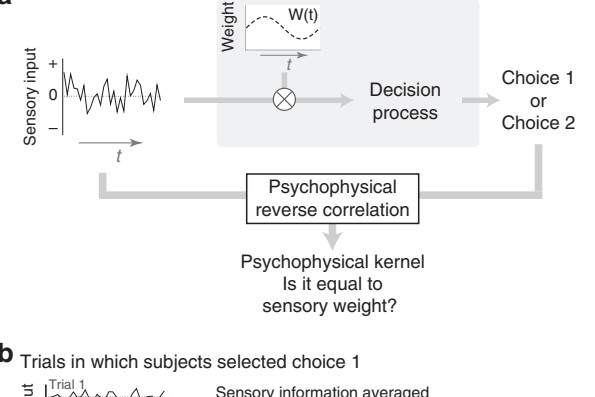

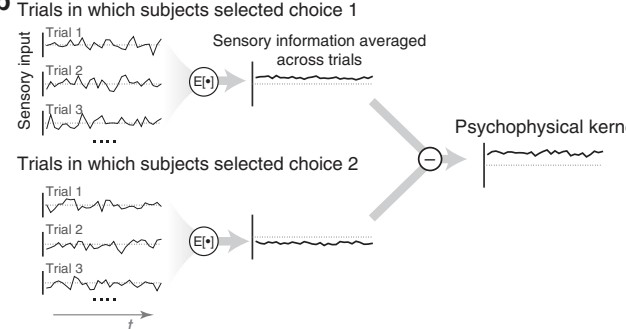

**Fig. 1** Psychophysical reverse correlation has been developed to recover sensory weights in perceptual tasks but it could also be influenced by decision-making mechanisms. **a** In a typical reverse correlation experiment, subjects receive a sequence of randomly fluctuating sensory information and make a binary choice. Experimenters can directly observe the stimulus and choice, but not the sensory weights and decision-making process (gray box). When choices are made by applying a sensory filter (weighting function) to the stimulus and comparing the result against a criterion, as proposed by signal detection theory, psychophysical reverse correlation will recover the sensory filter. However, it is unknown how well the analysis generalizes to more complex decision-making mechanisms. **b** Reverse correlation calculates the average stimuli preceding each choice and subtracts the results for the two choices. The outcome is a "psychophysical kernel."

correct for them, when possible, and prevents false conclusions about temporal variation of sensory weights. We demonstrate this point in a series of experiments by showing that the mechanism that underlies decisions predicts temporal dynamics of psychophysical kernels and quantitatively explains experimentally derived kernels.

## Results

**Reverse correlation deviates from true sensory weights.** In a typical reverse correlation experiment, subjects observe a sequence of noisy sensory stimuli and try to detect the presence of a target or categorize a stimulus[3,27–30] (Fig. 1). The stimuli could be a random dot kinematogram[26,27], oriented gratings or bars[4,31], or any other sensory inputs that randomly vary within or across trials along one or more stimulus attributes. The reverse correlation analysis calculates the relationship between subjects' choice and stimulus fluctuations by averaging over the stimuli that precede a particular choice. For two-alternative decision tasks, the analysis yields two kernels, one for each choice. Because of symmetry of the two choices, the kernels tend to be mirror images of each other[27,32]. Therefore, it is customary to subtract the two kernels and report the result (Fig. 1b):

$$K(t) = E[s(t)|\text{choice1}] - E[s(t)|\text{choice2}], \qquad (1)$$

where $E[s(t)|\text{choice1}]$ indicates the trial average of the stimulus at time $t$ conditional on choice 1, $s(t)$ is the stimulus drawn from a stochastic function with symmetric noise (e.g., Gaussian), and $K(t)$ is the magnitude of the psychophysical kernel at time $t$.

Psychophysical kernels are guaranteed to match the sensory filters when decisions are made by applying a static nonlinearity[6,7], for example, comparison to a decision criterion, as suggested by SDT[8]. However, recent advances suggest that SDT offers an incomplete characterization of the decision-making process. In particular, many perceptual decisions depend on integration of sensory information toward a decision bound[13–16,28,33,34], the decision bound can vary based on speed–accuracy tradeoff[17,18], the integration is influenced by urgency[35–37] and prior signals[14,33,38,39], and experimentally measured RTs consist of a combination of decision time and non-decision time[24,25].

A simple and commonly used class of decision-making models that takes these intricacies into account and provides a quantitative explanation of behavior in perceptual tasks is the drift diffusion model (DDM)[13–15] and its extensions[16,18–20,22,40].

In DDM, weighted sensory evidence is integrated over time until the integrated evidence (the decision variable, DV) reaches either an upper (positive) or a lower (negative) bound (Fig. 2), where each bound corresponds to one of the choices. We begin our exploration with the most basic model but will focus on more complex implementations later in the paper.

Neither the integration process nor the boundedness of the integration per se causes a systematic deviation of psychophysical kernels from true sensory weights. We define true sensory weights as the weights applied to the sensory stimulus to create the momentary evidence that will be accumulated over time for making a decision. In Methods, we provide the mathematical proof that in a simple DDM where decision bound and noise are constant over time and behavioral responses are generated as soon as the DV reaches one of the bounds (non-decision time = 0), psychophysical kernels are proportional to the sensory weights:

$$K(t) = \frac{2\sigma_s^2}{B} w(t), \qquad (2)$$

where $w(t)$ is the time-dependent weight, $\sigma_s^2$ is the variance of stimulus fluctuations, and $B$ is the height of the decision bound. Similar results can be obtained for unbounded DDMs (Eq. 14). Figure 3 shows simulations that confirm our proofs. Reverse correlation for an unbounded integration process with constant or sinusoidally varying weights recovers the true weighting function (Fig. 3a–c, Supplementary Fig. 1). Similarly, it yields the true weights for a bounded DDM (Fig. 3d–e, h), regardless of the decision-bound height.

Although the proportionality in Eq. 2 may suggest that psychophysical kernels can be successfully used to recover spatiotemporal dynamics of sensory weights, critical limitations prevent that in practice, as we explain below. The most common limitation is the experimenter's lack of knowledge about decision time, which is caused by asynchrony between the time that the DV reaches a decision bound (bound-crossing time or decision time) and the subject's report of the decision (when the choice becomes known to the experimenter). Such asynchronies stem from two sources: delays in neural circuitry and experimental design.

In many experiments, subjects are exposed to the stimulus for a duration determined by the experimenter and can report their choice only after a Go cue. In these "fixed-duration" designs, the exact decision time and its trial-to-trial variability are unknown to

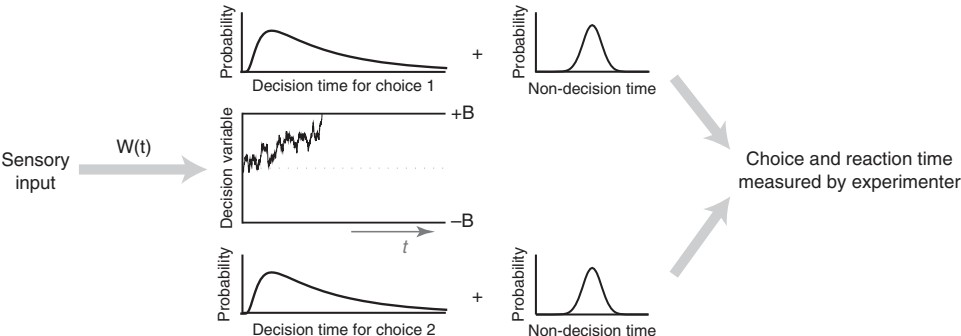

**Fig. 2** The drift diffusion model (DDM) captures the core computations for perceptual decisions made by integration of sensory information over time. We use variants of this model and more sophisticated extensions to explore how the decision-making mechanism influences psychophysical kernels. In DDMs, a weighting function, $w(t)$, is applied to the sensory inputs to generate the momentary evidence, which is integrated over time to form the decision variable (DV). The DV fluctuates over time due to changes in the sensory stimulus and neural noise for stimulus representation and integration. As soon as the DV reaches one of the two decision bounds ($+B$ for choice 1 and $-B$ for choice 2), the integration terminates and a choice is made (decision time). However, reporting the choice happens after a temporal gap due to sensory and motor delays (non-decision time). Experimenters know about the choice after this gap and can measure only the reaction time (the sum of decision and non-decision times) but not the decision time

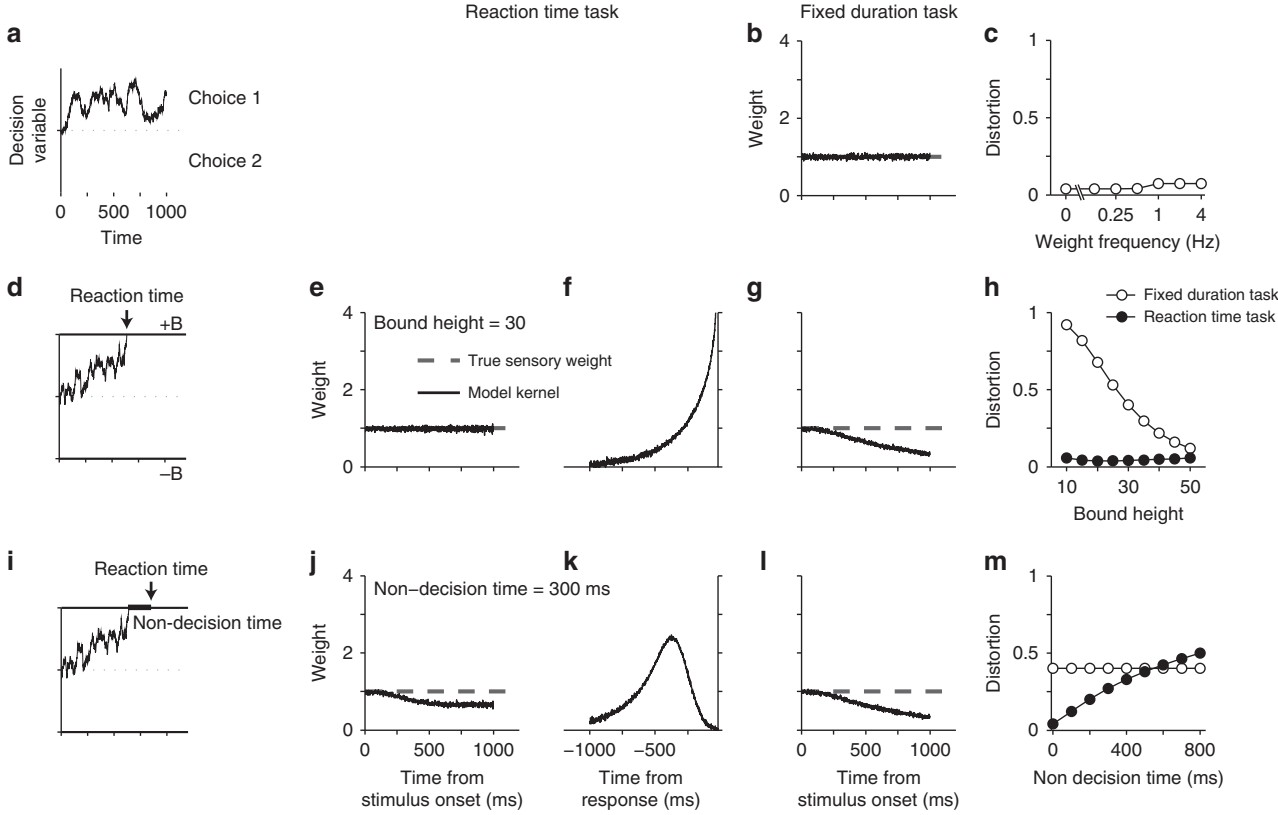

**Fig. 3** Psychophysical kernels deviate from sensory weights in DDM because of incomplete knowledge about decision time. **a–c** Integration of evidence per se does not preclude accurate recovery of sensory weights. For an unbounded DDM that integrates momentary evidence as long as sensory inputs are available, the kernel matches the true sensory weights. In this simulation, the weight is stationary and fixed at 1, but similarly matching results are obtained for any sensory weight (**c**; Supplementary Fig. 1). Distortion quantifies root-mean-square error between the psychophysical kernel and the true sensory weights (Eq. 17). **d–h** The decision bound does not preclude accurate recovery of sensory weights. In a bounded DDM without non-decision time, RTs are identical to decision time (**d**). Model simulations for RT tasks result in stimulus-aligned kernels that match sensory weights (**e**) and response-aligned kernels that rise monotonically (**f**), as expected for termination with bound crossing. However, stimulus-aligned kernels in fixed-duration tasks show a monotonic decrease because later stimuli are less likely to influence the choice (**g**). This deviation from true sensory weights is caused by early commitment to a choice and becomes smaller as the decision bound rises (**h**). **i–m** Variability of non-decision time makes reaction time an unreliable estimate of decision time, causing systematic deviations between psychophysical kernels and true sensory weights. After including non-decision time in the bounded DDM, stimulus-aligned kernels in RT tasks show a monotonic decrease because the stimuli that immediately precede the choice do not contribute to it (**j**). Response-aligned kernels show a peak, whose time is dependent on the distribution of non-decision times (**k**). Kernels for fixed-duration tasks are not affected by non-decision time (**l**) but still show the decline caused by bound crossing, similar to **g**. Deviation of stimulus-aligned kernels in the RT task increases with variability of non-decision time (**m**). Standard deviation of non-decision time is assumed to be 1/3 of its mean in these simulations. All kernels are normalized according to Eq. 2 or Eq. 14 to allow direct comparison with the true sensory weights (see Methods)

the experimenter, and decision times are likely to be prior to the Go cue[27,41]. Because stimuli presented after the bound-crossing time do not contribute to the choice (or contribute less), including that period in the calculation of psychophysical kernels leads to a progressive underestimation of sensory weights[26,27,30], causing a systematic deviation from Eq. 2 (Fig. 3g–h), compatible with past studies[42]. The diminishing kernel (Fig. 3g) correctly characterizes the effective reduction of the influence of the sensory stimulus on choice. However, note that from an experimenter's perspective, the shape of the kernel is inadequate to tell whether the reduced influence of the stimulus is caused by a change in sensory weights, by early termination of the decision during stimulus viewing, by a combination of both, or by another mechanism in the decision-making process (see below). Such a mechanistic understanding could be achieved only if the experimental design is enriched and a model-based approach is adopted. Although there are successful examples of achieving such goals[26,27], fixed-duration tasks impose significant limitations on experimenters' ability to determine the beginning and end of the decision-making process (cf. ref. [41]), which would be

necessary for separating sensory and decision-making mechanisms that shape psychophysical kernels.

Experimental designs in which subjects respond as soon as they make their decision (RT tasks; Fig. 3e) enable measurement of decision times and can be used to address the problem. However, RT tasks come with their own challenges. Sensory and motor delays are among them (Fig. 3i). Although the presence of such delays is widely appreciated, their effect on psychophysical kernels is unexplored. These delays effectively create a temporal gap between bound crossing and the report of the decision, making stimuli immediately before the report inconsequential for the decision. Figure 3j shows that non-decision times pull down the psychophysical kernel. These systematic reductions can cause the illusion of nonstationarity for stationary sensory weights (Fig. 3j, m) or distort the dynamics of time-varying weights (Supplementary Fig. 2).

What makes the psychophysical reverse correlation especially vulnerable to non-decision times is the variable nature of the sensory and motor delays[43,44]. A fixed non-decision time would cause a readily detectable signature (Supplementary Fig. 3) and is

easy to correct for by excluding the last stimulus fluctuation in each trial that corresponds to the non-decision time. Similarly, if the non-decision time was variable but we could know the exact delay on each trial, we could easily discard the corresponding period at the end of the stimulus to correct for the artificial dynamics caused by the non-decision time. In practice, however, the non-decision time is not a fixed number[25]. Further, the variability of non-decision time is often in the same order of magnitude as the decision time[20,34,45], making it challenging to thoroughly scrub away the effect of non-decision time just by trimming the stimuli. A more efficient solution is to embrace the distortion caused by the non-decision time, develop an explicit model of both sensory and decision-making mechanisms, and compare the model predictions with experimentally derived kernels (see the next section).

The fixed-duration design is not affected by the non-decision time, if there is a long enough delay between the stimulus and Go cue or if the stimulus duration is long enough to exceed the tail of the reaction time distribution in an equivalent RT task design (Fig. 3l–m). However, as mentioned above, lack of knowledge about the beginning and end of the integration process in fixed-duration tasks impedes mechanistic studies of kernel dynamics.

So far, we have focused on psychophysical kernels aligned to the stimulus onset. In an RT task, the stimulus-viewing duration varies from trial to trial and we can choose to align the kernel to subjects' responses. Such an alignment is informative both about the termination mechanism of the decision-making process and about the distribution of non-decision times. When the decision-making process stops by reaching a decision bound, the kernel is guaranteed to show a steep rise close to the decision time (Fig. 3f) because stopping is conditional on a stimulus fluctuation that takes the DV beyond the bound. This rise of the kernel does not indicate an increase of sensory weights immediately before the decision. Further, the magnitude of this rise is not always fixed and depends on the decision bound and distribution of non-decision times (see below; Supplementary Fig. 3). In the presence of a variable non-decision time (Fig. 3k), response-aligned kernels peak and then drop down to zero before the response. The drop happens because the non-decision time causes later fluctuations in the stimulus not to bear on the choice. The difference between the peak of the kernel and the reaction time is dependent on the mean and standard deviation of the non-decision time distribution, as well as its higher moments (Supplementary Fig. 3). Since it is known that the distribution of non-decision times can be quite diverse[46], the shape of the response-aligned psychophysical kernels can provide an important clue about the distribution of non-decision times and also verification of model-based attempts to discover the non-decision time distribution[46]. Overall, psychophysical kernels aligned to the response are influenced by sensory weights, termination criterion of the decision, and the non-decision time.

**Experimental measurements confirm model predictions**. The results of the previous section suggest that psychophysical kernels reflect a mixture of sensory and decision-making processes. By embracing this complexity, one can leverage psychophysical kernels to gain insight about both processes. The key is to develop explicit models and compare model predictions against experimentally derived kernels. Below, we highlight two experiments designed to achieve this goal.

The first experiment is an RT version of the direction discrimination task[20,35]. On each trial, subjects viewed a random dot stimulus and made a saccadic eye movement to one of the two targets as soon as they were ready to report their choice (Fig. 4a). Consistent with previous studies, accuracy improved

and RTs decreased monotonically with motion strength (Fig. 4b–c)[20,35]. We quantified moment-to-moment fluctuations of motion in each trial by calculating the motion energy[27,47,48] (see Methods; Supplementary Fig. 4). Figure 4d shows the average and standard deviation of motion energies across all 0% coherence trials and four single-trial examples. As expected, single-trial motion energies departed from 0 with a short latency[47] and then fluctuated between positive and negative values, which corresponded to the two motion directions discriminated by subjects. Across all 0% coherence trials, these fluctuations canceled each other out, resulting in a zero mean but the standard deviation remained large, indicating short bouts of varying motion strengths in either direction throughout the trial. The stochastic nature of the stimulus and the known effect of motion energy on the choice[27,48,49] provided an excellent opportunity to quantify how stimulus dynamics shaped the behavior.

Experimentally derived kernels for 0% coherence trials (Fig. 4e–f, red lines) showed a clear nonstationarity with remarkable resemblance to the kernels expected from a DDM with non-decision time and stationary sensory weights (Fig. 3j–k; the delayed rise of the psychophysical kernel in Fig. 4e is inherent to the motion energy calculation, as shown in Fig. 4d). We quantitatively tested the hypothesis that kernel dynamics reflect bound crossing and non-decision time by fitting the DDM to subjects' choices and RTs and generating a model prediction for the psychophysical kernels. Consistent with past studies, the distribution of RTs and choices across trials provided adequate constraints for estimating all model parameters[20,35], evidenced by the quantitative match between subjects' accuracy and RTs with model fits (data points vs. solid gray lines in Fig. 4b–c; $R^2$, 0.97 ± 0.01 for accuracy and 0.98 ± 0.01 for RTs, mean ± s.e.m. across subjects). After estimating the model parameters, we used them to predict the shape of the psychophysical kernel for the 0% coherence motion energies used in the experiment. These predicted kernels (Fig. 4e–f, solid gray lines) closely matched the experimentally derived ones ($R^2$, 0.57), establishing that the dynamics of the kernels were both qualitatively and quantitatively compatible with stationary sensory weights and a decision-making process based on bounded accumulation of evidence.

In a second experiment, we focused on a more complex sensory decision that required combining multiple spatial features over time (Fig. 5a). Subjects categorized faces based on their similarity to two prototypes. Each face was designed to have only three informative features (eye, nose, and mouth) (Fig. 5b). On each trial, the mean strengths (percent morph) of these three features were similar and randomly chosen from a fixed set spanning the morph line between the two prototypes. However, the three features fluctuated independently along their respective morph lines every 106.7 ms (Fig. 5c). All other parts of the faces remained fixed halfway between the two prototypes and, therefore, were uninformative. Further, each frame of the face stimulus was quickly masked to prevent conscious perception of fluctuations in eyes, nose, and mouth. Subjects reported the identity of the face (closer to prototype 1 or 2) with a saccade to one of the two targets, as soon as they were ready. The key difference with the direction discrimination task was that instead of one stimulus attribute that fluctuated over time (motion energy), there were three attributes that fluctuated independently. The three informative features could support the same or different choices in each stimulus frame and across frames. This task provided a richer setting to test how humans combine multiple spatial features to make a decision.

Consistent with the simpler direction discrimination task, as the average morph level of the three features approached one of the prototypes, choices became both more accurate and faster

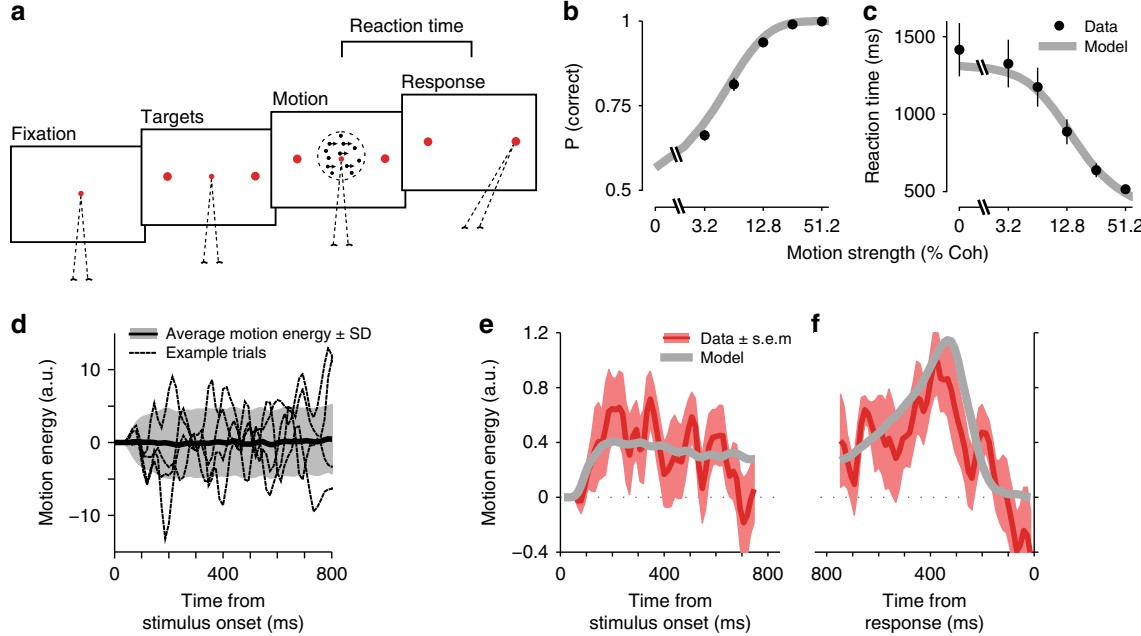

**Fig. 4** Psychophysical kernels in the direction discrimination task match predictions of a bounded DDM with non-decision time. **a** RT task design. Subjects initiated each trial by fixating on a central fixation point. Two targets appeared after a short delay, followed by the random dot stimulus. When ready, subjects indicated their perceived motion direction with a saccadic eye movement to a choice target. The net motion strength (coherence) varied from trial to trial, but also fluctuated within trials due to the stochastic nature of the stimulus. **b, c** Choice accuracy increased and RTs decreased with motion strength. Data points are averages across 13 subjects. Accuracy for 0% motion coherence is 0.5 by design and therefore not shown. Gray lines are fits of a bounded DDM with non-decision time. Error bars denote s.e.m. across subjects. **d** Motion energy of example 0% coherence trials (dotted lines), and the average (solid black line) and standard deviation (shading) of motion energy across all 0% coherence trials. Positive and negative motion energies indicate the two opposite motion directions in the task. **e, f** The bounded DDM predicts psychophysical kernels (gray lines), which accurately match the dynamics of subjects' kernels (red lines). Because the model sensory weights are stationary, kernel dynamics in the model are caused by the decision-making process and non-decision times. Kernels are calculated for 0% coherence trials. Shading indicates s.e.m. across subjects. All kernels are shown up to the minimum of the median RTs across subjects

(Fig. 5d–e). The psychophysical kernels of the three features (Fig. 5g) had rich dynamics. First, the eye kernels had larger amplitude than the mouth and nose kernels, suggesting that choices were more strongly influenced by fluctuations in the eye region[50,51]. Second, the stimulus-aligned kernels dropped gradually over time, and the saccade-aligned kernels showed a characteristic peak a few hundreds of milliseconds prior to the choice. A multi-feature integration process with stationary weights for eyes, nose, and mouth regions could quantitatively explain our results. For each stimulus frame, the model calculated a weighted sum of the three features to estimate the momentary sensory evidence and then integrated this momentary evidence over time in a bounded diffusion model (Fig. 5f, see Methods). Fitting the model to the choice and RT distributions provided a quantitative match for both (Fig. 5d–e; $R^2$, $0.998 \pm 0.001$ for accuracy and $0.98 \pm 0.01$ for RTs) and the resulting parameters led to kernels that well matched the dynamics of experimentally observed kernels for the three features ($R^2$, 0.74).

**Testing for temporal dynamics of sensory weights**. Our exploration of the model and fits to experimental data in the previous sections focused largely on cases in which sensory weights were static and the dynamics of the psychophysical kernel were solely due to the decision-making process. However, as discussed earlier, changes of sensory weights could also be a major factor in shaping psychophysical kernels (Fig. 3c, Supplementary Fig. 1 and 2). In theory, a model-based approach to understanding kernel dynamics should be able to distinguish changes of sensory weights from decision-making processes because of their distinct effects on the choice and RT

distributions. To test this prediction, we simulated a direction discrimination experiment in which decisions were made by accumulation of weighted sensory evidence toward a bound in the presence of non-decision time and various dynamics of sensory weights (Supplementary Fig. 5). Then, we used the simulated choice and RT distributions to fit an extended DDM that allowed temporal dynamics of sensory weights. The model recovered the weight dynamics and accurately predicted psychophysical kernels of the simulated experiments in each case (Supplementary Fig. 5). A few thousand trials, similar to those available in our experimental datasets, were adequate to achieve accurate fits and predictions. Therefore, there does not seem to be critical limitations in the ability of a model-based approach to detect sensory weight dynamics, when such dynamics are present.

Knowing about the model's ability, we extended the DDMs used in the previous section to explore dynamics of sensory weights for human subjects. The extended models included linear and quadratic terms to capture a wide variety of temporal dynamics (Eqs. 23 and 25). The results did not support substantial temporal dynamics of sensory weights in either task (12 out of 13 subjects of the direction discrimination task and all subjects of the face discrimination task showed static weights). Overall, the addition of temporal dynamics to the weight function did not significantly improve the fits or the match between model and experimental psychophysical kernels (for direction discrimination, Eq. 23, $\beta_1$, $-3.0 \pm 1.6$ across subjects, $p = 0.10$, median, $-0.65$, and $\beta_2$, $-2.1 \pm 2.3$, $p = 0.36$, median, 0.17; for face discrimination, Eq. 25, $\beta_1$, $-0.19 \pm 0.10$, $p = 0.09$, median, 0.10, and $\beta_2$, $0.055 \pm 0.028$, $p = 0.08$, median, 0.028). Because similar models could accurately recover weight dynamics in the

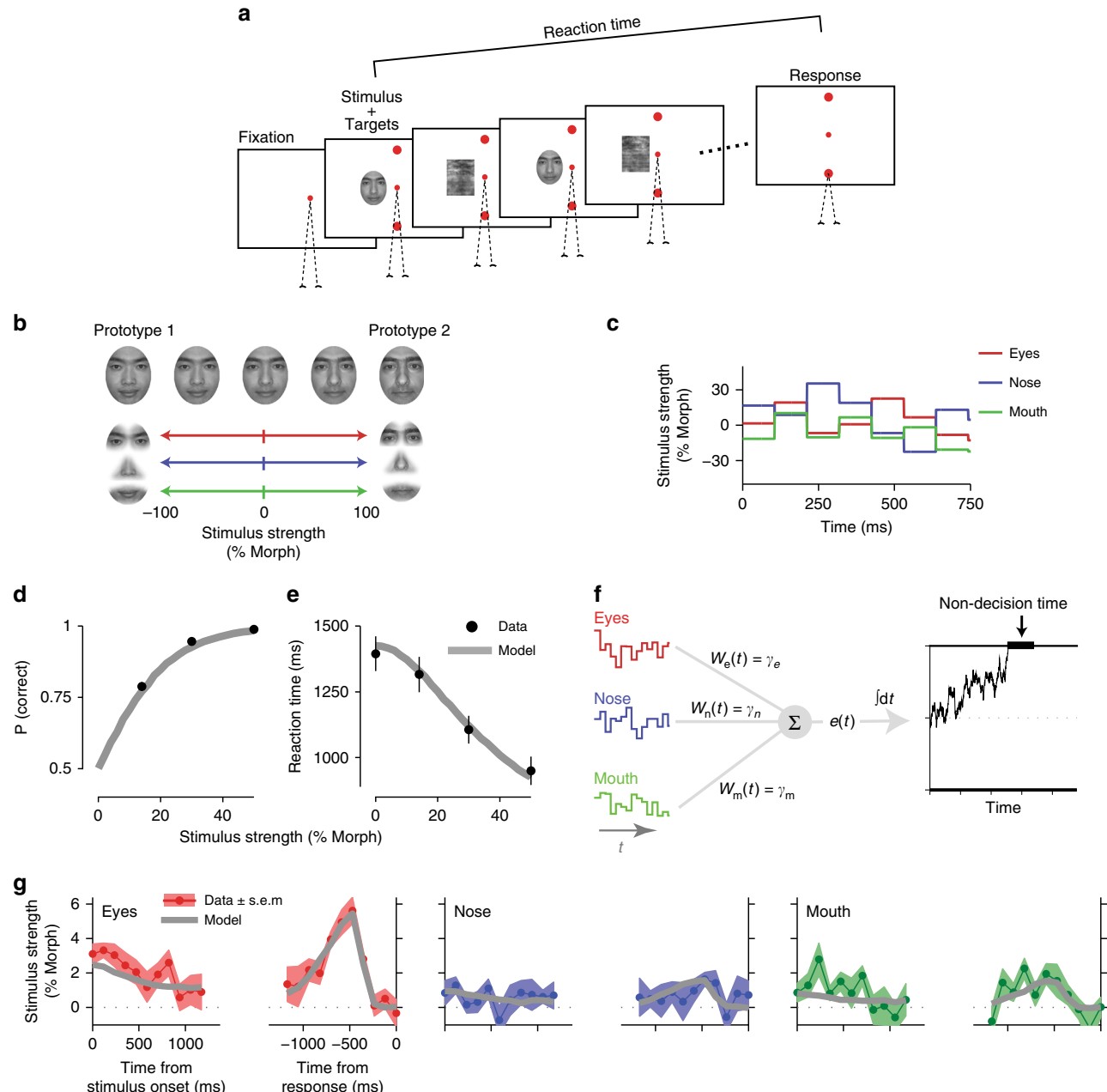

**Fig. 5** Psychophysical reverse correlation in a face discrimination task with multiple informative features reveals relative weighting of features and kernel dynamics similar to the direction discrimination task. **a** Task design. Subjects viewed a sequence of faces interleaved with masks and reported whether the face identity matched one of two prototypes. They reported their choice with a saccadic eye movement to one of the two targets, as soon as ready. **b** Using a custom algorithm, we designed intermediate morph images between the two prototype faces such that only three facial features (eyes, nose, and mouth) could be informative. These features were morphed independently from one prototype (+100% morph) to another (−100% morph), enabling us to create stimuli in which different features could be biased toward different identities. All regions outside the three informative features were set to halfway between the prototypes and were uninformative. **c** The three informative features underwent subliminal fluctuations within each trial (updated with 106.7-ms interval). The mean morph levels of the three features were similar but varied across trials. Fluctuations of the three features were independent (Gaussian distribution with standard deviations set to 20% morph level). **d**, **e** Choice accuracy increased and RTs decreased with stimulus strength. Data points are averages across nine subjects. Error bars are s.e.m. across subjects. Gray lines are model fits. **f** The DDM used to fit subjects' choices and RTs extends the model in Fig. 2 by assuming different sensitivity for the three informative features. Momentary evidence is a weighted average of three features where the weights correspond to the sensitivity parameters. The momentary evidence is integrated toward a decision bound. **g** Psychophysical kernels estimated from the model (gray lines) match subjects' kernels for the three features. Shaded areas are s.e.m. across subjects

simulated data, we do not think that our observation about the experimental data is caused by a low power for detection of weight dynamics or a fundamental bias to attribute changes of psychophysical kernels to the decision-making process.

**Speed–accuracy tradeoff, bias, and more complex decision models.** Although a simple DDM for accumulation of evidence captures several key aspects of behavior in sensory decisions, it is only an abstraction for the more complex computations

implemented by the decision-making circuitry. More complex and nuanced models are required both to explain details of behavior and to create biologically plausible models of integration in a network of neurons. We use this section to explore a non-exhaustive list of key parameters commonly used in various implementations of evidence integration models. For clarity, we simulate models without non-decision time to isolate the effects of these model parameters from those of non-decision time.

First, we focus on how changes of decision bound influence the shape of psychophysical kernels. The effect is best demonstrated by Eq. 2 for a simple DDM, which shows the kernel is inversely proportional to bound height. This dependence is expected because a lower decision bound boosts the effect of stimulus fluctuations on choice and vice versa. As a result, if subjects increase the decision bound to improve their accuracy[14,17,52], psychophysical kernels will shrink (Supplementary Fig. 6a–b).

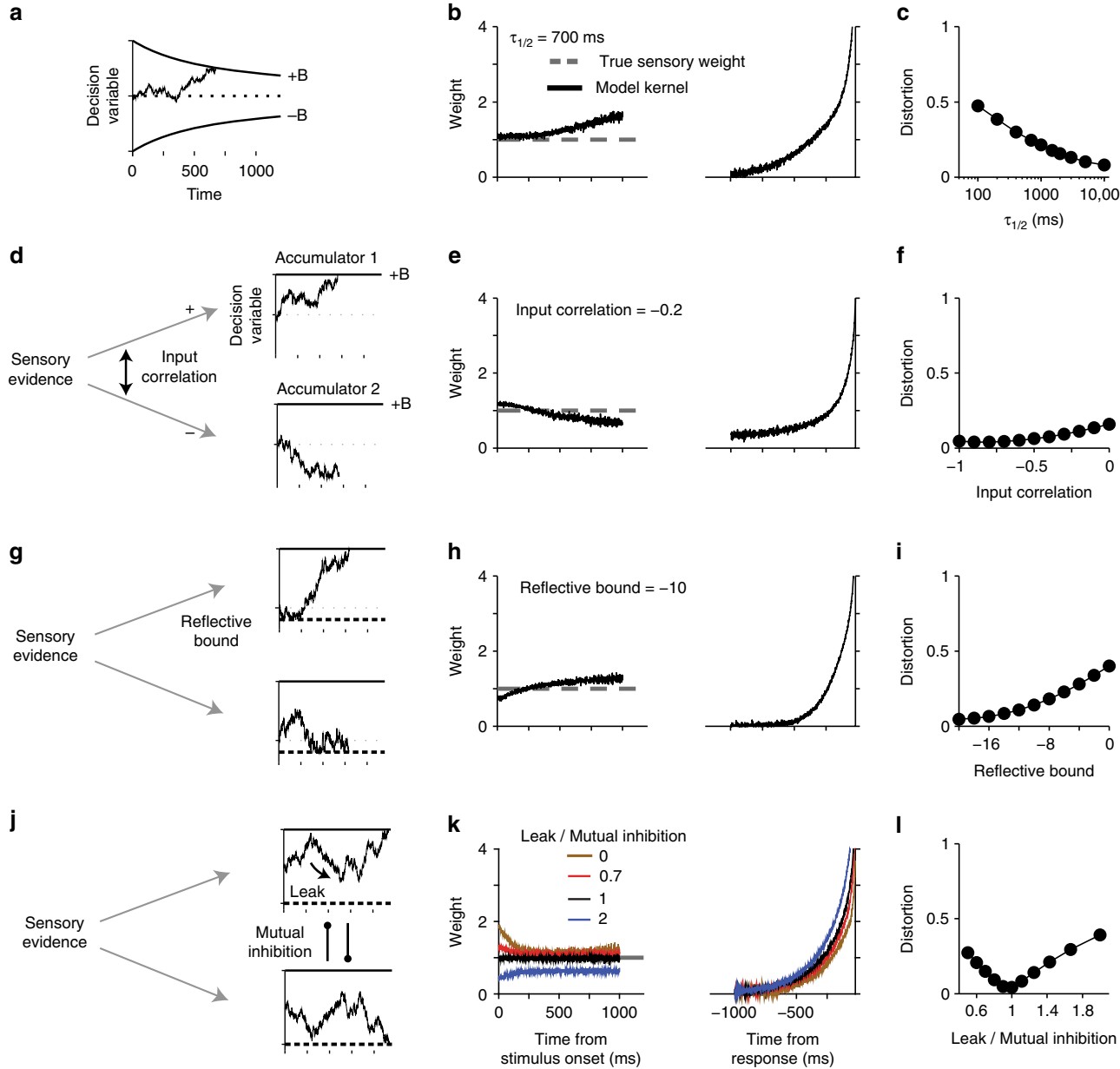

**Fig. 6** Psychophysical kernels are susceptible to changes of decision bound, input correlation, mutual inhibition, integration time constant, and limited dynamic range. The figure shows extensions of DDM and systematic deviations that additional realism to the model can cause in psychophysical kernels. Conventions are similar to Fig. 3, except that we focus only on RT tasks. Also, to isolate the effects of different model parameters from the effect of non-decision time, we use zero non-decision time in these simulations. **a–c** Collapsing decision bound (urgency signal) inflates the psychophysical kernel over time. The rate of bound collapse is defined by $\tau_{1/2}$—the time it takes to have a 50% drop in bound height. **d–f** Extending DDM to a competition between two bounded accumulators reveals that input correlation of the accumulators has only modest effects on psychophysical kernels, causing an initial overshoot followed by an undershoot compared to true sensory weights. **g–i** The presence of a lower reflective bound in the accumulators causes an opposite distortion: an initial undershoot followed by a later overshoot. **j–l** Balancing the effect of mutual inhibition by making the integrators leaky causes the model to behave like a DDM, eliminating the effects of both the inhibition and leak on the psychophysical kernels (black curves in m). Any imbalance between leak and inhibition, however, causes systematic deviations in the kernels from the true sensory weights (brown, red, and blue curves in **k**). See Supplementary Fig. 8 for more examples

Similarly, urgency signals, which push the integration process toward the decision bound[37,53], influence the kernels. Urgency is effectively a reduction of decision bound over time (Fig. 6a) and leads to inflation of the psychophysical kernel (Fig. 6b). The scaling of kernels with bound height can be largely corrected by estimating the decision bound from behavior and multiplying the kernels by it, as we did for the results in the previous sections.

The proportionality constant in Eq. 2 also points at another important conclusion: changes of stimulus variance, if present, systematically distort psychophysical kernels. Larger stimulus noise inflates the kernel and vice versa (Supplementary Fig. 6c–e). This contrasts with the effects of internal (neural) noise for the representation of sensory stimuli or the DV. We show in Methods that in a bounded DDM, internal noise does not have a systematic effect on psychophysical kernels of RT tasks (but compare to unbounded DDM).

The presence of choice bias in the decision-making process is another factor that can cause distortions in psychophysical kernels. Two competing hypotheses have been suggested for implementation of bias in the accumulation to bound models. One hypothesis is a static change in the starting point of the accumulation process (or an equivalent static change in decision bounds)[14,15,33,54], which would cause an initial inflation in the psychophysical kernels without a lasting effect (Supplementary Fig. 7a–c). A second hypothesis is a dynamic bias signal that pushes the DV toward one of the decision bounds and away from the other[38]. This dynamic bias signal can be approximated by a change in the drift rate of DDM, which would cause a DC offset in the psychophysical kernels (Supplementary Fig. 7d–f).

Electrophysiological recordings from motor-planning regions of the primate brain suggest that integration of sensory evidence is best explained with an array of accumulators, rather than a single integration process[37,40,55–57]. A class of models that matches this observation better than the simple DDM is competing integrators—one for each choice—that accumulate evidence toward a bound[16,19,20,23,40,57]. Our mathematical proof for DDM does not exactly apply to these models. However, many of these models can be formulated as extensions of the DDM with new parameters added to provide more flexible dynamics[23]. The following parameters are worth special attention: input correlation, lower reflective bound, mutual inhibition, and leak (see Supplementary Notes for more detailed explanations).

A DDM is mathematically equivalent to two integrators that receive perfectly anti-correlated inputs (correlation = −1) and, consequently, are anti-correlated with each other[20,23]. However, perfect anti-correlation in neural responses is not expected because even when signal correlations are negative, noise correlations tend to be close to zero or slightly positive[58,59]. Figure 6d–f shows that the shape of the psychophysical kernel is only minimally affected by a wide range of input correlations. Sizeable distortions arise only when the input correlation approaches 0, in which case the kernel is initially inflated but later drops below the true sensory weight (Fig. 6e and Supplementary Fig. 8a).

A frequent feature of biologically plausible implementations of the integration process is a lower reflective bound that limits the lowest possible DV[19–22,40]. Such reflective bounds are inspired by the observation that the spike count of neurons is limited from below and cannot become negative. Reflective bounds cause the psychophysical kernel to begin lower than the true sensory weight but exceed it later (Fig. 6g–i and Supplementary Fig. 8b). However, these distortions are small when the reflective bounds are far enough from the starting point of the integrators.

Several models incorporate mutual inhibition either through direct interactions between the integrators[19,40] or indirectly through intermediate inhibitory units[22,60]. Mutual inhibition is often combined with decay (leak) in the integration process (Fig. 6j) to create richer dynamics and curtail the effects of inhibition[19,23]. The balance between leak and mutual inhibition defines whether the model implements bistable point attractor dynamics or line attractor dynamics[23]. This balance also determines the kernel dynamics (Fig. 6j–l and Supplementary Fig. 8c). When mutual inhibition dominates (leak/inhibition ratio < 1), psychophysical kernels show an early amplification but later converge on the true sensory weights. When leak and inhibition balance each other out, the model acts similarly to a line attractor and the psychophysical kernels resemble those of a DDM. Finally, when leak dominates, the integrators lose information and psychophysical kernels systematically underestimate the sensory weights, especially for earlier sensory evidence in the trial.

Interestingly, and perhaps by luck, applying these more complex model variations to our experimental data resulted in model parameters that closely resembled linear integration of evidence, which is why the DDMs in the last section performed so well. Because of this parameterization, these more sophisticated models would produce predictions similar to the simple DDM about the dynamics of the psychophysical kernel. However, we note that this observation may not generalize to other experiments and should, therefore, be tested for new behavioral paradigms on a case-by-case basis.

As explained above, different parameters of decision-making models have different and even opposing effects on the expected shape of psychophysical kernels. As a result, a mixture of these features can, in principle, generate a variety of kernel dynamics, depending on their exact parameters. To illustrate this point, we consider models with two competing integrators that have different levels of mutual inhibition, leak, collapsing bounds, and sensory and motor delays (Fig. 7a). For static sensory weights over time, this class of models can generate monotonically decreasing kernels (Fig. 7b), monotonically increasing kernels (Fig. 7c), or kernels that exactly match the true sensory weights (Fig. 7d), depending on the model parameterization. To understand this diversity, consider, for examplem the opposing effects of collapsing bounds (urgency) and non-decision time on the kernels. The gradual reduction of the kernel due to non-decision time can cancel out the increase of the kernel due to urgency. Alternatively, one of the two effects may overpower the other one. Complementary to the examples in Fig. 7, one can also imagine parameterizations that would result in a flat psychophysical kernel in the presence of nonstationary true sensory weights. The presence of mutual inhibition and leak further complicates the relationship of sensory weights and psychophysical kernels and expands the space of possible dynamics for the kernels.

## Discussion

A key goal of systems neuroscience is to explain behavior as a sequence of neural computations that transform sensory inputs to appropriate motor outputs. For perceptual decisions, this sequence includes sensory processes that form neural representations of a stimulus in sensory cortices and decision-making processes that plan the best choice based on these sensory representations. Psychophysical reverse correlation has been originally developed to infer sensory filters that approximate sensory processes[3–5]. Recent studies, however, have begun to use the technique for inferring the properties of the decision-making process[26,27,61,62]. Here, we show that an isolated perspective is vulnerable. To ensure correct interpretation of psychophysical kernels, one has to adopt an integrative perspective that sees psychophysical kernels as a product of both sensory and decision-making processes.

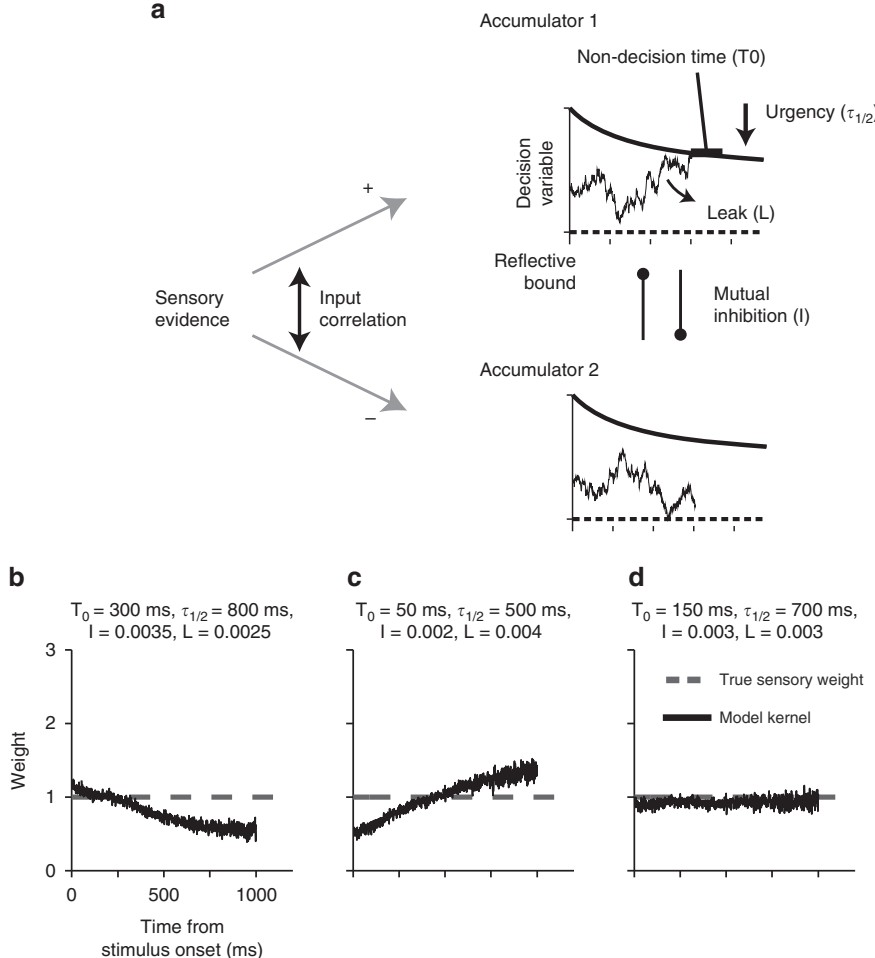

Fig. 7 A decision-making model that has a mixture of parameters with opposing effects on psychophysical kernels can create a diversity of kernel dynamics for static sensory weights. **a** A model composed of two competing integrators that allows different ratios of leak and inhibition, collapsing decision bounds, and non-decision times. The model also has input correlation >−1 and reflective lower bounds, but they are fixed for simplicity. **b** When bound collapse is small and non-decision times are long, the kernel drops monotonically over time. **c** When bound collapse is large and non-decision times are short, the kernel rises monotonically. **d** When these opposing factors balance each other, the kernel becomes flat

We arrived at our conclusion through a systematic exploration of how psychophysical reverse correlation is influenced by the decision-making process and under what conditions it provides a good approximation of sensory filters. We showed that neither the integration of sensory evidence nor the termination of the integration process by reaching a decision bound fundamentally limits the recovery of sensory weights. However, nuances that are fixtures of real experiments but often receive little attention can be major sources of deviation between psychophysical kernels and sensory weights. Examples include sensory and motor delays or input correlation of competing integrators, which cause kernels with a downward trend unrelated to sensory weights, or urgency and lower reflective bounds, which cause upward trends in the kernels. Previous theoretical explanations of reverse correlation have ignored these nuances, causing confusion about what can be gleaned from psychophysical kernels. We also showed that the kernels are susceptible to how the integration process is implemented. Bistable point attractor or line attractor dynamics, implemented through different combinations of mutual inhibition, self-excitation, and decay of activity (leak) in neural networks, yield different kernel dynamics. We conclude that psychophysical kernels are influenced by both sensory and decision-making processes and show how they can be used to provide information about both types of processes.

Making the interpretation of psychophysical reverse correlation dependent on the decision-making process is likely to face opposition because of the historical influence of SDT[8] and the fact that under the assumptions of SDT, reverse correlation matches the true sensory weights. However, we note that this match is misleading. In particular, SDT explains the dynamics of psychophysical kernels by simplifying the decision-making process and shifting its complexity to the sensory processes. This shift is inaccurate both because it depends on unsubstantiated sensory processes and because it ignores the known complexity in the decision-making process. The perspective offered by SDT is also insufficient because it fails to explain changes of psychophysical kernels that stem from flexibility of the decision-making process. For example, setting the speed and accuracy of choices[17,18] or adjustment of behavior following feedback[35,63] often depends on rapid alterations in the decision-making process. These alterations also change psychophysical kernels, as explained in Results. The integrative framework proposed here would correctly identify the source of changes, whereas SDT would have to misattribute them to changes of sensory weights.

Proper partitioning of the contribution of sensory and decision-making processes enables testing hypotheses about neural mechanisms of behavior. We provide key signatures of psychophysical kernels for a variety of different mechanisms and

implementations. Matching these signatures to the patterns observed in an experiment enables hypothesis formation. To quantitatively test those hypotheses, one can implement computational models of sensory and decision-making processes, fit them to some aspect of behavior, and then generate predictions about another aspect of behavior. For example, by fitting the DDM to the distribution of choices and RTs, we could accurately predict the dynamics of psychophysical kernels in our experiments. Further, our mathematical proofs and simulations provide a comprehensive framework for predicting changes of psychophysical kernels under various experimental manipulations. For example, when a manipulation leads to improved accuracy, one can use our method to distinguish two potential sources of the improved performance: increased sensitivity (e.g., attentional mechanisms) or changes of decision bound (speed–accuracy tradeoff). Whereas increased sensitivity is expected to increase the magnitude of the psychophysical kernel, increased decision bound or reduced urgency is expected to reduce the kernel magnitude. Such contrasting predictions highlight the ability of psychophysical reverse correlation to separate models that may not be easily distinguished by more conventional measurements, including changes in psychometric function or its derivatives such as overall accuracy.

However, psychophysical kernels on their own are inadequate to determine the nuances of sensory and decision-making processes. Multiple mechanisms influence the reverse correlation and mixtures of mechanisms can generate complex dynamics or even flat kernels, complicating model-free interpretations of experimentally derived kernels. (See Supplementary Discussion.) To reduce interpretational errors, one should always assess psychophysical kernels in conjunction with the choice and RT distributions. Mechanisms that have a similar effect on psychophysical kernels often have contrasting effects on choices and RTs. For example, variability of non-decision times, input correlation, and mutual inhibition can all lead to stimulus-aligned kernels that shrink with time. However, as shown in Supplementary Fig. 9a, these three mechanisms (i) have different effects on the shape of the tail of RT distribution, (ii) make different predictions about accuracy for the same sensitivity function, and (iii) differ in the exact shape of the psychophysical kernels (note the small but detectable, qualitative and quantitative differences in the dynamics of the kernels). The combination of these signatures enables separation of different mechanisms. A similar contrast in kernel dynamics and RT and choice distributions exists for dropping bounds and reflective bounds, which both cause an inflation of psychophysical kernels over time. Along the same lines, when combinations of different mechanisms with opposing effects lead to a flat kernel (e.g., Fig. 7d), the distribution of RT and choice is often different compared to when the kernel is flat because none of those mechanisms contribute to the decision-making process or a different combination of parameters flattens the kernel (Supplementary Fig. 9b). A three-pronged approach based on the shape of the kernel, and RT and choice distributions makes a powerful technique for uncovering the mechanisms that shape behavior.

In light of our results, past studies that relied on psychophysical reverse correlation can teach us more than most of them were designed for. It is still valid to interpret psychophysical kernels as the best approximation of a linear–nonlinear model, such as SDT, to the behavior. However, it is important to keep in mind that such an interpretation is about effective associations between sensory information and choice[64], which should not be confused with the spatiotemporal filters that shape sensory representations, or the readout of sensory representations to form the evidence used in the decision-making process. It is also important to note that these effective associations do change

under various conditions that do not change sensory representations (e.g., changes of decision bound), while the model-based approach suggested here is likely to recover the true dynamics of sensory weights. The results of our paper do not refute careful use of psychophysical reverse correlation in past studies. Rather, we try to elevate psychophysical reverse correlation from a technique that reveals only effective associations of stimulus and choice to a technique that reveals the inner working of the sensory and decision-making processes that underlie the choice.

## Methods

**Overview**. We examine how well psychophysical reverse correlation recovers sensory weights in perceptual tasks where decisions are based on accumulation of sensory information[13–15,19]. First, we prove that the core computations for integration of evidence or termination of the decision-making process on based on a decision bound do not cause any deviation of psychophysical kernels from true sensory weights. Then, we demonstrate that non-decision times, urgency, reflective bounds, and interactions between accumulators are substantial sources of deviation. Finally, we show how one can accurately interpret the complexity of psychophysical kernels by explicit modeling of both sensory and decision-making processes.

**Psychophysical reverse correlation for bounded accumulation**. DDMs are commonly used to approximate integration of evidence in two-alternative sensory decisions[13–15]. In these models, a weighting function is applied to momentary sensory evidence and the result is integrated over time until the integrated evidence (the DV) reaches either an upper (positive) or a lower (negative) bound. Each bound corresponds to one of the choices. The sensory weighting function is assumed constant in some experiments such as direction discrimination of random dots where sensory neurons show more or less constant activity proportional to stimulus strength throughout the stimulus presentation period[65]. However, the exact form of the weighting function is usually unknown in most experiments and could change dynamically depending on context. We prove below that in the absence of sensory and motor delays, reverse correlation accurately recovers the weights applied to sensory evidence in a DDM.

In a reaction time task, where subjects report their choices as soon as ready, the psychophysical kernel is

$$K(t) = E\big[s(t)|C^1_{T \geq t}\big] - E\big[s(t)|C^2_{T \geq t}\big], \tag{3}$$

where $C^i_{T \geq t}$ indicates all trials in which choice $i$ is made at times equal or larger than $t$, and $E\big[s(t)|C^i_{T \geq t}\big]$ is the average stimulus at time $t$ conditional on the choice being made at a later time ($T \geq t$). The intuition for this formulation is that a trial contributes to the calculation of the kernel at time $t$ only if the choice on that trial is not recorded by the experimenter before $t$. For an unbiased decision maker and a stimulus distribution symmetric around zero, $E\big[s(t)|C^1_{T \geq t}\big] = -E\big[s(t)|C^2_{T \geq t}\big]$, leading to $K(t) = 2E\big[s(t)|C^1_{T \geq t}\big]$. Therefore, we need to calculate only one of the two conditional averages.

In a DDM, the choice is made through integration of sensory evidence:

$$v_i(t) = \int_0^t \big[w(\tau)s_i(\tau) + \eta_i(\tau)\big]\mathrm{d}\tau, \tag{4}$$

where $v_i(t)$ indicates the DV on trial $i$ at time $t$, $w(\tau)$ is the weight applied on the stimulus at times $\tau \leq t$, $s_i(\tau)$ are stimuli sampled from a Gaussian distribution with mean 0 and variance $\sigma_s^2$, and $\eta_i(\tau)$ represents internal (neural) noise for the representation of sensory and integration processes. We assume that the internal noise does not bias the representation and can be approximated with a Gaussian distribution with mean 0 and variance $\sigma_\eta^2$.

Because integration continues until $v_i(t)$ reaches one of the two bounds ($+B$ or $-B$),

$$\begin{aligned} E\big[s(t)|C^1_{T \geq t}\big] &= \sum_i s_i(t)p\big(s_i(t)|C^1_{T \geq t}\big) \\ &= \sum_i s_i(t) \int \int_D p\big(s_i(t)|v_i(t), \eta_i(t), C^1_{T \geq t}\big) p\big(v_i(t), \eta_i(t)|C^1_{T \geq t}\big)\mathrm{d}v\mathrm{d}\eta, \end{aligned}$$

$$\tag{5}$$

where the integration domain $D$ is $[-B,+B]$. Using Bayes rule and by plugging Eq. 5 in Eq. 3, we get

$$K(t) = \frac{2}{p\big(C^1_{T \geq t}\big)} \sum_i s_i(t)p(s_i(t)) \int \int_D p\big(C^1_{T \geq t}|s_i(t), v_i(t), \eta_i(t)\big) p\big(v_i(t), \eta_i(t)\big)\mathrm{d}v\mathrm{d}\eta,$$

$$\tag{6}$$

where $p(C^1_{T \geq t}|s_i(t), v_i(t), \eta_i(t))$ is the probability of reaching the upper bound after time $t$, when the decision maker observes stimulus $s_i(t)$ and has an existing decision variable $v_i(t)$. This bound-crossing probability has an analytical solution in DDM[15]. Note that after observing stimulus $s_i(t)$, the distance of the accumulated evidence from the upper bound is $\chi_1 = B - (w(t)s_i(t) + v_i(t) + \eta_i(t))$, and the distance from the lower bound is $\chi_2 = B + (w(t)s_i(t) + v_i(t) + \eta_i(t))$. Because $s(t)$ and $\eta(t)$ have zero mean, the overall drift is zero and the bound-crossing probability is

$$p(C^1_{T \geq t}|s_i(t), v_i(t), \eta_i(t)) = \frac{\chi_2}{\chi_2 + \chi_1}$$

$$= \frac{B + (w(t)s_i(t) + v_i(t) + \eta_i(t))}{2B}. \tag{7}$$

Therefore, Eq. 6 can be written as

$$K(t) = \frac{1}{Bp(C^1_{T \geq t})} \sum_i s_i(t)p(s_i(t)) \left[ \begin{array}{c} (B + w(t)s_i(t)) \iint_D p(v_i(t), \eta_i(t)) \mathrm{d}v\mathrm{d}\eta + \\ \iint_D (v_i(t) + \eta_i(t))p(v_i(t), \eta_i(t)) \mathrm{d}v\mathrm{d}\eta \end{array} \right]$$

$$= \frac{p(C_{T \geq t})}{Bp(C^1_{T \geq t})} \sum_i s_i(t)p(s_i(t))(B + s_i(t)w(t)), \tag{8}$$

where $p(C_{T \geq t})$ is the combined probability of reaching either of the two decision bounds after time $t$. The second equality in Eq. 8 stems from two equations. First,

$$\iint_D (v_i(t) + \eta_i(t))p(v_i(t), \eta_i(t)) \mathrm{d}v\mathrm{d}\eta = 0 \tag{9}$$

because for a neutral stimulus, the DV is symmetrically distributed around the starting point. Second,

$$\iint_D p(v_i(t), \eta_i(t)) \mathrm{d}v\mathrm{d}\eta = p(C_{T \geq t}) \tag{10}$$

because $\int_{-B}^{+B} p(v_i(t)) \mathrm{d}v$ reflects the total probability of the DV between the decision bounds at $t$, and because this unabsorbed probability mass is guaranteed to be fully absorbed by the decision bounds in finite time $T \geq t$[66].

For an unbiased decision-making process and a stimulus with zero mean $p(C_{T \geq t}) = 2p(C^1_{T \geq t})$. As a result, Eq. 8 simplifies to

$$K(t) = \frac{2}{B} \left[ B \sum_i s_i(t)p(s_i(t)) + w(t) \sum_i s_i(t)^2 p(s_i(t)) \right]. \tag{11}$$

For a Gaussian stimulus with zero mean, $\sum_i s_i(t)p(s_i(t)) = 0$ and $\sum_i s_i(t)^2 p(s_i(t)) = \sigma_s^2$. Therefore,

$$K(t) = \frac{2\sigma_s^2}{B} w(t)$$

This equation, which we highlight in Results (Eq. 2), shows that the result of psychophysical reverse correlation is proportional to the sensory weights. The proportionality constant is $\frac{2\sigma_s^2}{B}$, which explains how reverse correlation is modulated by properties of the stimulus (stimulus variance) and parameters of the decision-making process (decision bound). Eq. 2 also shows that psychophysical kernels are independent of internal noise in a DDM. Internal noise does not cause a systematic deviation in estimated kernels, although it could affect the confidence interval of the estimated kernels in real experiments, where a limited number of trials are available for measuring the kernels.

Based on Eq. 2, the outcome of psychophysical reverse correlation is expected to change if the decision bound is not constant. For example, urgency in the decision-making process is often equivalent to a drop in the decision bound[35,37,53], which should lead to a gradual increase of the reverse correlation kernel even in the absence of changes in sensory weights (Fig. 6a–c). Note that we define urgency as an additive signal for competing accumulation processes, which under certain conditions (e.g., anti-correlated input to the accumulators) can be translated to collapsing bounds in the DDM. This is different from the alternative definition of urgency based on gain in the accumulation process[36,45], which cannot be easily translated to a bound change in DDM.

The proof above holds only if the choice can be recorded as soon as the decision-making process terminates. In practice, one has to take into account sensory and motor delays that postpone initiation of action. These delays imply that the stimuli presented immediately before the behavioral response do not influence the choice[49]. As a result, the psychophysical kernel drops to zero prior to the choice. Due to trial-to-trial variability of these delays, it is not possible to know purely based on behavior, which part of the stimulus did or did not contribute on any individual trial, but on average, one can expect a descending trend in a psychophysical kernel close to the time of the response (Fig. 3i–m).

In addition to the bound height and non-decision time, other factors can cause deviation of psychophysical kernels from sensory weights. DDM is a simplified model of the more complex computations implemented by the neural circuit that underlies the choice. In the simplest case, one should consider an array of accumulators that interact and compete with each other[22,26,40], forcing us to consider correlation between accumulators, mutual inhibition, and leak which can cause systematic deviations in the kernels (Figs. 6d–f, 6j–l). Further, real neurons do not accommodate negative firing rates. A lower reflective bound in each accumulator can introduce additional systematic biases in the kernel (Fig. 6g–i). A closed-form, mathematical solution for the psychophysical kernel in the presence of all these factors is complex and beyond the scope of this paper. Therefore, we use simulations to explore the parameter space of different model variations and demonstrate how different factors change the kernel (see "Model simulation" below).

**Psychophysical reverse correlation for unbounded accumulation**. If decisions in a fixed-duration task are made by unbounded integration of evidence, psychophysical kernels will correctly reflect the dynamics of sensory weights (Fig. 3a–c, Supplementary Fig. 1). The proof is as follows. If integration begins with stimulus onset and continues for the whole stimulus duration, $T_s$, the DV at the end of the stimulus in trial $i$ will be

$$v_i(T_s) = \int_0^{T_s} \left[ w(\tau)s_i(\tau) + \eta_i(\tau) \right] \mathrm{d}\tau. \tag{12}$$

If the sensory input, $s(t)$, is drawn from a Gaussian distribution with mean 0 and variance $\sigma_s^2$, $v(T_s)$ will have a mean of 0 and variance $\sigma_{tot}^2 = T_s \sigma_\eta^2 + \sigma_s^2 \int_0^{T_s} w(\tau)^2 \mathrm{d}\tau$. The model selects choice 1 for the positive DV and choice 2 for the negative DV. Therefore, the psychophysical kernel will be

$$\begin{aligned} K(t) &= E[s(t)|v(T)>0] - E[s(t)|v(T)<0] \\ &= 2E[s(t)|v(T_s)>0] \\ &= 2 \sum_i s_i(t)p(s_i(t)|v(T_s)>0) \\ &= \frac{2}{p(v(T_s)>0)} \sum_i s_i(t)p(s_i(t))p(v(T_s)>0|s_i(t)) \\ &= \frac{2}{p(v(T_s)>0)} \sum_i s_i(t)p(s_i(t)) \left[ \int_0^{+\infty} N(x, |w(t)s_i(t), \sigma_{tot}) \mathrm{d}x \right], \\ &= 4 \sum_i s_i(t)p(s_i(t))\Phi(w(t)s_i(t)/\sigma_{tot}) \\ &= 4 \int_{-\infty}^{+\infty} sp(s)\Phi(w(t)s/\sigma_{tot}) \mathrm{d}s \end{aligned} \tag{13}$$

where $\Phi(x)$ is the cumulative distribution function of a standard normal probability density function with mean 0 and standard deviation 1. The last equality in the equation is due to the i.i.d. property of $s(t)$ within and across trials.

The kernel equation can be further simplified as

$$\begin{aligned} K(t) &= 4 \int_{-\infty}^{+\infty} sp(s)\Phi(w(t)s/\sigma_{tot}) \mathrm{d}s \\ &= \frac{4\sigma_s^2 w(t)}{\sqrt{2\pi[w(t)^2\sigma_s^2 + \sigma_{tot}^2]}} \\ &\approx \frac{4\sigma_s^2}{\sqrt{2\pi}\sigma_{tot}} w(t), \end{aligned} \tag{14}$$

where the approximation in the last line is based on $w(t)^2\sigma_s^2 \ll \sigma_{tot}^2$, which is usually true unless stimulus durations are very short.

Based on Eq. 14, the kernel is proportional to the sensory weight function, and the constant of proportionality scales with stimulus variance $(\sigma_s^2)$, similar to Eq. 2 for bounded accumulation.

However, note that the kernel is inversely proportional to $\sigma_{tot}$, which is a function of the stimulus duration. Dependence of kernels on stimulus duration calls for caution in interpretation of results when a mixture of stimulus durations are used in an experiment. The kernel for each stimulus duration is scaled differently, inducing artificial dynamics in the average kernel across all durations.

Even when stimulus durations are the same across trials, subjects can begin integration at variable times across trials and commit to a choice at different times during stimulus viewing, causing temporal dynamics in the kernel that do not reflect the true dynamics of sensory weights (see Results and Discussion for a more detailed explanation).

**Model simulation**. We simulated four different classes of bounded accumulation models: (i) DDM without non-decision time, (ii) DDM with non-decision time[14,15,24], (iii) DDM with non-decision time and urgency[18,35,37], and (iv) competing accumulators with different input correlation, reflective bound, mutual inhibition, and leak[16,19,20,23,40,57].

For each class and parameter combination, $10^6$ trials were simulated to obtain an accurate estimation of the psychophysical kernel. Sensory input for each trial was a sequence of independent draws from a Gaussian distribution with mean 0 and $\sigma_s = 1$. This input was multiplied with a weight function, $w(t)$, which could be constant (Figs. 3, 6, 7, Supplementary Fig. 3, 6–9) or vary over time (Supplementary Fig. 1–2, 5). This weight function dictated the significance that each sample played in shaping the decision. The outcome (termed momentary evidence) was passed to each integration model to calculate the DV and generate a choice for each trial. As explained above, the presence or absence of internal noise did not bear strongly on the measured kernels in RT tasks, as long as enough trials were available for the measurement. We calculated psychophysical kernels based on the simulated choices and sensory stimuli according to Eq. 3 and compared the results against the weight function used in the simulation (Figs. 3, 6, 7, Supplementary Fig. 1–2, 6–9).

**Drift diffusion models.** In these models, momentary evidence was integrated over time until the DV reached a lower bound ($-B$) or an upper bound ($+B$), which corresponded to the two choices. For models without non-decision time, integration stopped immediately and a choice was registered after the bound crossing. For models with non-decision time, the integration process stopped after bound crossing but the choice was registered after a random time, drawn from a Gaussian distribution. The stimuli presented between bound crossing and the choice were included in the calculation of psychophysical kernels to emulate realistic experimental conditions, where experimenters do not know the exact non-decision time on each trial. The mean and standard deviation of the distribution of non-decision time in Fig. 3j–l and Supplementary Fig. 2 were 300 ms and 100 ms, respectively, compatible with past studies[24,34,35,45]. In Fig. 3m, the mean varied between 0 and 800 ms, and the standard deviation was equal to 1/3 of the mean. In Supplementary Fig. 3, we tested the effects of different means, variance, and skewness of the non-decision time distribution on the measured psychophysical kernels.

For DDMs with urgency, we reduced the decision bound according to a hyperbolic function:

$$B(t) = b - u_\infty \frac{t}{t + \tau_{1/2}}, \tag{15}$$

where $b$ is the initial bound height, $u_\infty$ is the asymptotic reduction in bound height, and $\tau_{1/2}$ is the time to reach 50% of the reduction. We set $b = 60$, $u_\infty = 60$, and $\tau_{1/2} = 400$ ms for Fig. 6b. In Fig. 6c, $\tau_{1/2}$ varied while the other model parameters were kept constant.

For the simulation of fixed-duration tasks (Figs. 3b, g, l, Supplementary Fig. 1), we incorporated past experimental observations that the decision-making process could effectively stop before the termination of the stimulus[27,41]. Stimulus durations were 1 s on all trials. The full stimulus duration was used for the calculation of the psychophysical kernel to reflect the standard practice and experimenters' lack of knowledge about the exact time of the decision on each trial.

**Competing accumulator models.** DDM is a low-parameter model and by design lacks the sophistication of a biologically plausible neural network that implements the integration process[22,67,68]. A more biologically plausibility alternative is a model with a bank of accumulators (integrators) that interact and compete with each other. For a two-alternative decision, the simplest instantiation of such a model has two accumulators, each integrating sensory evidence in favor of one of the two choices[19,20]. The model reaches a decision when one of the accumulators crosses its bound. In addition to the DDM parameters (bound height and non-decision time), the competing accumulator model has the following parameters (see Eq. 16):

1. Input correlation ($\rho$) determines the correlation between sensory inputs of the two accumulators. The inputs are explained by a two-dimensional Gaussian distribution with mean 0 and covariance matrix $\psi = \begin{pmatrix} 1 & \rho \\ \rho & 1 \end{pmatrix} \sigma_e^2$, where $\sigma_e^2$ reflects the combined variance of weighted stimulus noise and internal noise.

2. The second parameter is a reflective bound ($R$) that defines a lower limit for the DV of each accumulator.

3. The third parameter is the strength of inhibitory interactions between the accumulators ($I$). This mutual inhibition is widely assumed to be a key component of biological circuits of decision-making and a key factor in shaping neuronal response dynamics[19,22]. When $I > 0$, the strength of mutual inhibition for accumulators 1 and 2 at time $t$ is $Iv_2(t)$ and $Iv_1(t)$, respectively, where $v_1$ and $v_2$ are the DVs of the two accumulators. Because the magnitude of inhibition is proportional to the accumulated evidence, even small $I$ can have dramatic effects on the decision-making process.

4. The fourth model parameter is "leakage" in the integration process ($L$). In the absence of mutual inhibition, the leak makes the model behave as an Ornstein–Uhlenbeck process[66], causing the DVs to decay faster as they get farther from their starting point. In the presence of mutual inhibition, the balance of leak and inhibition creates a variety of attractor dynamics. When the leak and inhibition parameters are equal ($L = I$), the difference of the DVs of the two accumulators implements a DDM: a line attractor that reflects the accumulated difference of

momentary evidence of the two accumulators[23]. When mutual inhibition exceeds leak ($L < I$), a saddle point emerges in the state space of the model, which exponentially amplifies small initial differences of the DVs of the two accumulators over time. This amplification boosts the effect of early stimulus fluctuations on the decision[19]. Conversely, when the leak parameter exceeds mutual inhibition ($L > I$), a point attractor emerges in the state space, causing differences in the DVs of the two accumulators to decay over time. This decay reduces the effect of early stimulus fluctuations on the choice.

The equation that governs our simulations of competing accumulator models in Figs. 6, 7, and Supplementary Fig. 8 is

$$d\begin{bmatrix} v_1 \\ v_2 \end{bmatrix} = \left( w(t)S(t) - L\begin{bmatrix} v_1 \\ v_2 \end{bmatrix} - I\begin{bmatrix} v_2 \\ v_1 \end{bmatrix} + v_0(L+I) \right)dt + dW \tag{16}$$
$$v_1(0) = v_2(0) = v_0,$$

where $dv$ denotes the change in $v$ over a small time interval $dt$, $L$ is the leak term, $I$ is the mutual inhibition, and $v_0$ is the starting point of DVs. $S(t)$ is a vector that represents the sensory inputs to the two accumulators. For the simulations in Figs. 6, 7, and Supplementary Fig. 8, we assumed that two accumulators were driven in opposite directions by the input stimuli, that is $S(t) = \begin{bmatrix} s(t) \\ -s(t) \end{bmatrix}$. $dW$ is a 2D Gaussian noise term with mean 0 and covariance $\xi dt$, where $\xi = \begin{pmatrix} 1 & \rho' \\ \rho' & 1 \end{pmatrix} \sigma_\eta^2$. We adjusted $\rho'$ to achieve a desired input correlation ($\rho$) as defined above. $v_1$ and $v_2$ started at $v_0$. $v_0(L+I)$ created a stable parameter at $v_0$. The DVs were subjected to two nonlinearities: a lower reflective bound ($R$) and an upper absorbing bound ($B$).

Figure 6e–f demonstrates distortions in the psychophysical kernels for different input correlations ($\rho$ was set to $-0.2$ in Fig. 6e and varied between $-1$ and 0 in Fig. 6f; $B = 30$, $R = -\infty$, $I = 0$, $L = 0$, $\sigma_\eta^2 = 1$, and $v_0 = 0$). In the absence of a lower reflective bound, inhibition, or leak, the model became mathematically equivalent to a DDM whenever $\rho = -1$. In Fig. 6h–i, we tested the effect of a lower reflective bound ($R$ was set to $-10$ in Fig. 6h and varied between $-20$ and 0 in Fig. 6i; $\rho = -1$, $B = 30$, $I = 0$, $L = 0$, $\sigma_\eta^2 = 0$, and $v_0 = 0$). Figure 6k–l shows how the balance of leak and mutual inhibition distorted psychophysical kernels. For these simulations, we kept $L+I = 0.006$ and systematically changed the ratio $L/I$ between 0.5 and 2 ($\rho = -1$, $B = 60$, $R = 0$, $\sigma_\eta^2 = 0$, and $v_0 = 30$). We also show the shape of the kernel in the absence of a leak (brown lines, $I$ is set to 0.003). For these simulations, the lower reflective bound was set to 0 to ensure that negative DVs in one accumulator did not excite the other accumulator. To best isolate the effect of individual parameters of the model, we set the non-decision time and urgency to zero in Fig. 6d–l. Figure 7 shows the effect that conjunctions of different parameters have on the psychophysical kernel. The standard deviation of non-decision time was set to 1/3 of its mean in this figure.

**Comparison of model kernels and sensory weights.** For each model, we calculated the psychophysical kernel as explained by Eq. 3. To directly compare the kernels with the sensory weights implemented in the model, we divided the kernels by the scaling factor of Eq. 2 $\left( \frac{2\sigma_s^2}{B} \right)$. For models with dynamic bounds (Figs. 6a–c, 7), we used the average bound height from the stimulus onset to the median RT to calculate the scaling factor. For unbounded models (Fig. 3a–c), we used the scaling factor in Eq. 14 $\left( \frac{4\sigma_s^2}{\sqrt{2\pi}\sigma_{tot}} \right)$. After scaling the kernels and making them comparable to the sensory weights, we quantified the difference between the stimulus-aligned weight and kernel functions using root-mean-square error:

$$\text{distortion} = \sqrt{\frac{1}{T_m} \sum_{t=1}^{T_m} (w(t) - K(t))^2}, \tag{17}$$

where $T_m$ is the stimulus duration in simulations of fixed-duration tasks or the median RT in simulations of RT tasks.

**Overview of psychophysical tests.** We performed two experiments to test our model predictions: direction discrimination with random dots, and a novel face discrimination task. All subjects were naïve to the purpose of the experiments and provided informed written consent before participation. All procedures were approved by the Institutional Review Board at New York University. Throughout the experiments, subjects were seated in an adjustable chair in a semi-dark room with chin and forehead supported before a CRT display monitor (refresh rate 75 Hz, viewing distance 52–57 cm). Stimulus presentation was controlled with Psychophysics Toolbox[69] and Matlab. Eye movements were monitored using a high-speed infrared camera (Eyelink, SR-Research, Ontario). Gaze positions were recorded at 1 kHz.

**Direction discrimination task.** Thirteen human subjects performed an RT version of the direction discrimination task with random dots[20,24,35]. Data from six subjects have been previously reported in Kiani et al.[20] and data from the remaining subjects have been reported in Purcell and Kiani[35]. Both studies used a similar trial structure. Subjects initiated each trial by fixating a small red point at the center of

the screen (FP, 0.3° diameter). After a variable delay, two targets appeared on the screen, indicating the two possible motion directions. Following another random delay, the dynamic random dots stimulus appeared within a 5–7° circular aperture centered on the FP. The stimulus consisted of three independent sets of moving dots shown in consecutive frames. Each set of dots was shown for one video frame and then replotted three video frames later ($\Delta t = 40$ ms; density, 16.7 dots/deg$^2$/s). When replotted, a subset of dots were offset from their original location (speed, 5°/ s), while the remaining dots were placed randomly within the aperture. The percentage of coherently displaced dots determined the strength of motion. On each trial, motion strength was randomly chosen from one of six possible values: 0%, 3.2%, 6.4%, 12.8%, 25.6%, and 51.2% coherence. Subjects reported their perceived direction of motion with a saccadic eye movement to the choice target in the direction of motion. Once the motion stimulus appeared, subjects were free to indicate their choice at any time. RT was recorded as the difference between the time of motion onset and eye movement initiation. For the calculation of psychophysical kernels, motion energy of the random dot stimulus was calculated for each trial over time (see below).

**Face discrimination task**. Nine human subjects performed a novel experiment designed to test our model predictions for more complex decisions on multi-dimensional sensory stimuli. Subjects reported the identity of a face on each trial as soon as they were ready (Fig. 5a). The stimuli in the experiment were drawn from morph continuums between photographs of faces from the MacBrain Face Stimulus Set[70] (http://www.macbrain.org/resources.htm). For the illustrations in Fig. 5, we used morphed images of two of the authors to avoid copyright issues. We developed a custom algorithm that morphed different facial features (regions of the stimulus) independently between two prototype faces. Our algorithm started with 97 manually defined anchor points on each face and morphed one face into another by linear interpolation of the positions of anchor points and textures inside the tessellated triangles defined by the anchor points. The result was a perceptually seamless transformation of the geometry and internal features from one face to another. The anchor points also enabled us to morph different regions of the faces independently. We focused on three key features (eyes, nose, and mouth) and created independent series of morphs for them. The faces that were used in the task were composed of different morph levels of these three features. Anything outside those features was set to the halfway morph between the prototypes. The informativeness of the three features (stimulus strength) was defined based on the mixture of prototypes, spanning from −100% when the feature was identical to prototype 1 to +100% when it was identical to prototype 2 (Fig. 5b). At the middle of the morph line (0% morph), the feature was equally shaped by the two prototypes.

By varying the three features independently, we could study spatial integration through creating ambiguous stimuli in which different features could support different choices. We could also study temporal integration of features by varying the three discriminating features within each trial (Fig. 5c). The three discriminating features for each stimulus frame were drawn from independent Gaussian distributions. The mean and standard deviation of these distributions were equal and fixed within each trial, but the means varied randomly from trial to trial. We tested seven mean stimulus strengths (−50, −30, −14, 0, +14, +30, and +50% morph level). The standard deviation was 20% morph level. Sampled values that fell outside the range [−100% + 100%] (0.18% of samples) were replaced with new samples inside the range.

Changes of the stimulus within a trial were implemented in a subliminal fashion such that subjects could not consciously perceive variation of facial features and yet their choices were influenced by these variations. We achieved this goal using a sequence of stimuli and masks within each trial. The stimuli were morphed faces with a particular combination of the three discriminating features. The masks were created by phase randomization of the intermediate face between the two prototypes. For the majority of subjects (7/9), each stimulus was shown without a mask for one monitor frame (13.3 ms). Then, it gradually faded out over the next seven frames as a mask stimulus faded in. For these frames, the mask and the stimulus were linearly combined, pixel-by-pixel, according to a half-cosine function, such that in the last frame, the weight of the mask was 1 and the weight of the stimulus was 0. Immediately afterward, a new stimulus frame with a new combination of informative features was shown, followed by another cycle of masking, and so on. For a minority of subjects (2/9), we replaced the half-cosine function for the transition of a stimulus and mask with a full cosine function, where each eight-frame cycle started with a mask, transitioned to an unmasked stimulus in frame 5, and transitioned back to a full mask by the beginning of the next cycle. We did not observe any noticeable difference in the results of the two presentation methods and pooled their data. The masks ensured that subjects did not perceive minor changes in key features over time within a trial. In debriefings following each experiment, all subjects noted that they saw one face in each trial but the face was covered with various masks over time.

**Analysis of behavioral data**. Due to the stochastic nature of the random dot motion stimuli, the strength of motion of a stimulus with a fixed coherence fluctuated from one frame to another. We quantified these stimulus fluctuations by calculating motion energy[47]. Details are described elsewhere[27]. Briefly, we used two pairs of spatiotemporal filters, each selective for one of the two motion directions

discriminated by the subject. Each direction-selective filter was formed by summation of two space–time separable filters. The spatial filters were even and odd symmetric fourth-order Cauchy functions:

$$f_1(x,y) = \cos^4(\alpha)\cos(4\alpha)\exp\left(-\frac{y^2}{2\omega_g^2}\right),$$
$$f_2(x,y) = \cos^4(\alpha)\sin(4\alpha)\exp\left(-\frac{y^2}{2\omega_g^2}\right),$$

(18)

where $\alpha = \tan^{-1}(x/0.35)$ and $\omega_g = 0.05$. The two temporal filters were

$$g_1(t) = (60t)^3\exp(-60t)\left[\frac{1}{3!} - \frac{(60t)^2}{(3+2)!}\right],$$
$$g_2(t) = (60t)^5\exp(-60t)\left[\frac{1}{5!} - \frac{(60t)^2}{(5+2)!}\right].$$

(19)

The two pairs of direction-selective filter were constructed by combining the two spatial filters with the two temporal filters: $f_1 g_1 + f_2 g_2$ and $f_2 g_1 - f_1 g_2$ selective for one motion direction, whereas $f_1 g_1 - f_2 g_2$ and $f_2 g_1 + f_1 g_2$ were selective for the opposite direction. The parameters of Eq. 18 and Eq. 19 were chosen to (i) match spatial and temporal band-pass properties of MT neurons, (ii) to maximize selectivity of the directional filters for the speed of coherent motion in the stimulus (5°/s), and (iii) to reproduce the width of direction-selectivity tuning curves of MT neurons. We convolved the 3D spatiotemporal pattern of the stimulus in each trial with these four filters, squared the results, and then summed them for each pair of filters to measure local motion energies at each stimulus subregion over time. The local energies were summated across space and subtracted from the energy of the opposing pair of filters to obtain fluctuations of the net motion energy in one direction over time.

Average motion energies increased linearly with stimulus coherence (Supplementary Fig. 4b). However, the lag in the temporal filters caused the effect of stimulus fluctuations to show up in the motion energies with ~50-ms delay (Fig. 4d and Supplementary Fig. 4a), as shown before[27,47].

For the direction discrimination task, we used motion energies of 0% coherence trials to perform reverse correlation (Eq. 3) on the responses of human subjects (3389 trials). In Fig. 4e–f, we first computed each subject's psychophysical kernel and then averaged the kernels across subjects. Each subject's stimulus- and response-aligned kernels were calculated up to the subject's median RT, ensuring that at least half of trials contributed to the calculations. When averaged across subjects, the kernels were shown up to the shortest median RT. For the response-aligned kernels (Fig. 4f), we rounded the RT to the onset of the last stimulus frame on the monitor. The temporal resolution of kernels (13.3 ms) was dictated by the refresh rate of the monitor (75 Hz).

For the face discrimination task, we used fluctuations of eyes, nose, and mouth morph levels in the 0% morph trials to calculate psychophysical kernels of individual features (Fig. 5c) (3530 trials). Similar conventions to the direction discrimination task were used for averaging kernels across subjects and plotting them, except that because of the longer stimulus frame durations in the face discrimination task (106.7 ms), the kernels were temporally coarser. We did not perform any smoothing of the psychophysical kernels of the two tasks to avoid obscuring their dynamics.

**Fitting models to behavioral data and predicting psychophysical kernels**. We used a simple DDM to fit subjects' choices and RTs in the direction discrimination task in order to predict their psychophysical kernels. The model had four degrees of freedom: decision-bound height ($B$), mean non-decision time ($T_0$), standard deviation of non-decision time $\left(\sigma_{T_0}\right)$, and a sensitivity parameter ($\gamma$). The sensitivity parameter determined the mean of momentary evidence ($\mu = \gamma C$) conferred by a motion stimulus with coherence $C$. The bound height and sensitivity were in units of the standard deviation of the momentary evidence per unit time ($\sigma_e$), which we set to 1. This formulation of DDM, which has been used widely in the past, directly maps to the formulation presented earlier in Methods:

$$w(t) = \gamma,$$
$$\sigma_e^2 = \gamma^2\sigma_s^2 + \sigma_\eta^2.$$

(20)

The probability of crossing the upper and lower decision bounds at each decision time was calculated by solving the Fokker–Planck equation:[66]

$$\frac{\partial p(v,t)}{\partial t} = \left[-\frac{\partial}{\partial v}\mu + 0.5\frac{\partial^2}{\partial v^2}\sigma_e^2\right]p(v,t),$$

(21)

where $p(v,t)$ is the probability density of the DVs at different times. The boundary conditions were

$$p(v,0) = \delta(v),$$
$$p(\pm B, t) = 0,$$

(22)

where $\delta(v)$ denotes a delta function. The first condition enforced that the DV

always started at 0 and the second condition guaranteed that the accumulation terminated when the DV reached one of the bounds. RT distributions for each choice were obtained by convolving the distribution of bound crossing times with the distribution of non-decision times (Fig. 2).

We fit model parameters by maximizing the likelihood of the joint distribution of the observed choices and RTs in the experiment. For a set of parameters, the model predicted a distribution of RTs for each possible choice for the stimulus strength used in each trial. These distributions were used to calculate the log-likelihood of the observed choice and RT on single trials. These log-likelihoods were summed across trials to search for the best set of model parameters that maximized this sum. The model parameters were fit separately for each subject. To avoid local maxima, we repeated the fits from 10 random initial points and chose the parameters that maximized the likelihood function. Figure 4b–c shows the average fits across subjects. The high quality of fits for individual subjects and the average subject indicated that the DDM provided an adequate explanation for the computations underlying behavior in the direction discrimination task. Compatible with past studies, adding urgency to the model, or replacing the DDM with a competing accumulator model did not fundamentally change the fits because the parameterization of these more complex models stayed in a regime that approximated the line attractor dynamics of the DDM[20,23].

To test if a time-varying weighting function provided a better fit to the behavioral results, we modified Eq. 20 by adding linear and quadratic temporal modulations to the drift rate:

$$\mu(t) = \gamma \times (1 + \beta_1 t + \beta_2 t^2) \tag{23}$$

where $\beta_1$ and $\beta_2$ are additional degrees of freedom in the model.

We used the best model parameters that fit the RT and choice distributions to predict subjects' psychophysical kernels. We use the term prediction because moment-to-moment fluctuations of motion energies were not used to fit the model parameters and the fitting procedure did not create any explicit link between these fluctuations in the stimuli used in the experiments and single-trial choices and RTs. Using the model parameters, we predicted choices and RTs for $10^5$ simulated trials with 0% motion coherence and calculated motion energy kernels for the model choices. Because the sensitivity parameter of the model was calculated for motion coherence, we first divided motion energies by the slope of the line that related average motion coherence to the average motion energy (Supplementary Fig. 4). This division converted motion energy fluctuations within a trial into equivalent stimulus coherence fluctuations, which were directly passed to the model to generate a choice and a reaction time for each simulated trial. We used these choices and RTs to calculate the model prediction for the kernels and superimposed them on subjects' kernels (Fig. 4e–f).

For the face discrimination task, we extended the simple DDM explained above to include three different sensitivity parameters for the three facial features ($\gamma_e$ for eyes, $\gamma_n$ for nose, and $\gamma_m$ for mouth), increasing the total number of parameters to 6 (the other parameters were $B$, $T_0$, and $\sigma_{T_0}$). The mean momentary evidence at each time in a trial was

$$\mu(t) = \gamma_e s_e(t) + \gamma_n s_n(t) + \gamma_m s_m(t), \tag{24}$$

where $s_e(t)$, $s_n(t)$, and $s_m(t)$ were the morph levels of eyes, nose, and mouth at time $t$ on the trial. Note that $\mu(t)$ is a time-varying drift rate based on the exact fluctuations of stimulus strengths on individual trials, unlike the drift rate in the model for the direction discrimination task. Our goal was to obtain the relative sensitivity for the three informative facial features. Because the average morph levels of the three features were identical in each trial, using the average morph to derive the drift rate would have made the three sensitivity parameters redundant. The fitting procedure to subjects' choices and RTs was as explained for the direction discrimination task, except that we used Eq. 24 to include a time-varying drift rate, $\mu(t)$. Also, note that because we used the exact stimulus fluctuations in the Fokker–Plank equation, $\sigma_s$ was excluded from the definition of noise $\left(\sigma_e^2 = \sigma_\eta^2 = 1\right)$. The model parameters were fit separately for each subject using the maximum-likelihood procedure explained above.

To test if a time-varying weighting function provided a better fit to the behavioral results, we modified Eq. 24 to allow linear and quadratic temporal modulations to the drift rate, similar to what we did for Eq. 23:

$$\mu(t) = (\gamma_e s_e(t) + \gamma_n s_n(t) + \gamma_m s_m(t)) \times (1 + \beta_1 t + \beta_2 t^2) \tag{25}$$

The procedure for deriving the model kernels of the face discrimination task was similar to that for the direction discrimination task. We simulated $10^5$ trials with 0% morph and passed the fluctuations of the three informative features to get model choices and RTs for individual trials. We then used these choices to calculate the model kernels for the three features and superimposed the result on subjects' psychophysical kernels for comparison (Fig. 5g). Because we used the stimulus fluctuations for fitting the model parameters, the kernels derived from the model were not pure predictions, unlike the direction discrimination task. However, note that the model kernels were not directly fit to match the data either. They were calculated based on an independent set of simulated 0% morph trials, making the comparison in Fig. 5g informative.

**Code availability**. The custom code for data analysis and models is available from the corresponding author upon request.

**Data availability**. The data are available from the corresponding author upon request.

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

## Acknowledgements

The authors thank Michael Waskom and John Sakon for insightful comments on the manuscript. This work was supported by the National Institutes of Health Grant R01-MH109180, a Sloan Research fellowship, and a McKnight Scholar Award to R.K. L.S. was supported by NRSA training grant R90-1R90DA043849, B.A.P. was supported by a postdoctoral fellowship from the Simons Collaboration on the Global Brain, and G.O. was supported by postdoctoral fellowships from Japan Society for the Promotion of Science and from Charles H. Revson Foundation.

## Author contribution

G.O. and R.K. conceived the study, developed the theory, performed the simulations, analyzed the data, and wrote the paper. G.O., L.S., B.A.P., and R.K. conducted the experiments.

## Additional information

**Competing interests:** The authors declare no competing interests.

