## [Peer Review File · Nature Communications]

Reviewers' comments:

Reviewer #1 (Remarks to the Author):

Psychophysical reverse correlation is a commonly used method to infer the sensory weighting of stimulus information to inform a perceptual decision. Under assumptions of signal detection theory, reverse correlation has been shown to accurately recover true sensory weighting.

However, in this interesting and important article, the authors provide a strong case that reverse correlation does not provide an accurate recovery of true sensory weighting under a variety of common assumptions about decision-making in the sequential sampling framework.

Using a set of stimulations combined with empirical data, the authors show that various characteristics of decision-making distort the reverse correlation recovery of sensory weighting. These characteristics include, most importantly, non-decision time and non-decision time variability, between- and within-trial adjustment of response caution, and to a lesser extent also information leakage, mutual inhibition of accumulators, a reflective bound, or a non-perfect anti-correlation between input of accumulations.

The authors conclude that reverse correlation thus not only reflects sensory weighting, but also characteristics of decision-making. These distortions can be sufficiently strong to warrant serious caution in the interpretation of psychophysical kernels. In light of these findings, the authors illustrate how some previous findings using reverse correlations may need reinterpretation.

At the same time, the authors argue that various characteristics of decision-making aspects have key signatures in the psychophysical reverse correlation. Proper partitioning of the contribution of sensory and decision-making influences on the reverse correlation can still allow meaningful interpretation of reverse correlations.

Taken together, the authors adequately address an interesting research question. However, I have some remaining questions and comments. I am confident that the authors can address these and recommend publication pending major revision.

Major

1. In the Discussion (p. 16 L. 44 and further), the authors suggest that it is possible to disentangle the influences of sensory and decision-making characteristics, allowing for meaningful interpretation of psychophysical kernels. The reasoning seems to be that researchers could fit a decision-making model to reaction time and accuracy data (e.g., the DDM), use the fitted model to predict a psychophysical kernel, and compare the predicted kernel with the empirically obtained kernel. The key signatures, provided in the Results, could then be used to hypothesize which additional (decision-making) mechanisms took place.

However, I am not convinced of this approach for two reasons. First, the reasoning does not seem to allow the researcher to reject the assumption that sensory weighting is static over time: static sensory weighting is assumed by the DDM that is fitted, and any deviations between the predicted and empirically obtained kernels is interpreted in terms of decision-making effects other than sensory weighting (i.e., in terms of the effects of the decision-making characteristics covered in the Results / Figure 6). Could the use of reverse correlation then also falsify the assumption of static sensory weighting?

Second, this approach does not seem to consider the possibility of a mixture of decision-making mechanisms influencing the reverse correlation at the same time. The key signatures provided in Figure 6 are obtained when considering a single decision-making effect in isolation of all others. It remains an open question whether these 'key signatures' are still recognizable if multiple decision-making characteristics operate at the same time (e.g., non-decision time variability combined with a change in threshold). Illustrating this complexity, the authors show on P.14-15 and Figure 7 that combining multiple characteristics can lead to complex distortions in the psychophysical reverse correlation, even including the possibility that multiple decision-making effects balance each other out. Given the variability of effects and their similarity, are the effects of decision-making mechanisms on the reverse correlation identified?

Further adding to this point is that the data presented in Figure 6 are simulated with a high

number of trials (106). While this provides a clean/noise-free description of the key signatures, empirical data is noisy (e.g., Figure 4e). Therefore, it is not entirely clear how well the key signatures could be recovered in empirical data, even if they are to occur in isolation of any other influence on the kernels.

I think the article would improve from further clarification of this approach in the Discussion, or some evidence that this approach is feasible. Or otherwise, I wonder if the authors see any other methods that may be more suitable to explore the sensory weighting of stimuli to inform decisions?

Minor

1. P. 8, Figure 4b and P. 10, Figure 5d: In both plots, there seems to be a data point missing for stimulus/motion strength 0?
2. P. 4, L. 16-17: "One of the simplest and most commonly used decision-making models that takes these intricacies into account (...) is the drift diffusion model". The phrasing here seems to suggest that the (standard) DDM takes all of the 'intricacies' mentioned on lines 11-14 on the same page into account, but this is not the case for urgency. Maybe a slightly different phrasing can prevent confusion for readers not familiar with the DDM.
3. The simulations assume normally distributed non-decision times, and some variations. Usually, distributions of non-decision times are chosen based on mathematical simplicity rather than cognitive theory (e.g., the uniform distributions in the DDM, Ratcliff & Tuerlinckx (2002), and LBA, Brown & Heathcote (2008)). However, recent advances have made it possible to estimate the shape of the non-decision time distribution (Verdonck & Tuerlinckx, 2016), showing various shapes of non-decision time distributions. I feel this result is very interesting in light of the presented article and could provide the reader further intuitions on the interpretation of Figure S3 specifically.
4. P. 12 L2-3: "This urgency ... over time", and p.20, L22-23 "For example, ... decision bound". This mathematical equivalence depends on the exact dynamics of the decision bound (e.g., Hawkins et al., 2015).
5. P. 12, L.43: "Another commonly ... can go". As far as I'm aware (and the authors only cite) the Leaky, Competing Accumulator model as the only model that makes this assumption.
6. The key signatures provided in Figure 6 (and supplementary figure S5 and S6) give an overview of how various decision-making mechanisms/characteristics can distort reverse correlations. The mechanisms under study seem mainly chosen for their neural plausibility. Other decision-making characteristics that are often studied in behavioral experiments are neglected: bias/start point effects, and between-trial variability in start point and drift rate. Since the characteristics in the Results provide a sufficiently convincing case that distortions in reverse correlation can occur, I do not feel that all the effects of bias and bias/drift between-trial variability should be addressed in detail with simulations (which would provide a substantial amount of extra work). However, I wonder if the authors can shed some light in general terms of what could be expected from bias and between-trial variabilities? Or is there any reason to believe these characteristics may not be as important as the effects considered? (Since the proportionality constant is only valid assuming equal bounds (derivation Eqs. 3-11), all threshold effects on the reverse correlation would presumably be different when a bias is present?)
7. It seems that Figure 6j-l are special cases of Figure 6m-o, where mutual inhibition > leakage. In the two-accumulator version, the leaky competing accumulator model (Usher & McClelland, 2001, their Eqs. 5-6) reduces to an OU-process, where the difference between mutual inhibition and leakage determines the dynamics of the evidence accumulation process. Wouldn't it be simpler to illustrate both the effects of mutual inhibition and leakage with Figure 6m-o, and remove Figure 6j-l? Or does Figure 6j-l provide details not present in Figure 6m-o?

References

Brown, S.D., & Heathcote, A. (2008) The simplest complete model of choice reaction time: Linear

ballistic accumulation. *Cognitive Psychology*, 57, 153-178.

Hawkins, G. E., Forstmann, B. U., Wagenmakers, E.-J., Ratcliff, R., & Brown, S. D. (2015). Revisiting the Evidence of Collapsing Boundaries and Urgency Signals in Perceptual Decision-Making. *The Journal of Neuroscience*, 35, 2476-2484.

Ratcliff, R., & Tuerlinckx, F. (2002). Estimating parameters of the diffusion model: Approaches to dealing with contaminant reaction times and parameter variability. *Psychonomic Bulletin & Review*, 9(3), 438-481.

Usher, M. & McClelland, J. L. (2001). The time course of perceptual choice: The leaky, competing accumulator model. *Psychological Review*, 108(3), 550-592.

Verdonck, S., & Tuerlinckx, F. (2016), Factoring Out Nondecision Time in Choice Reaction Time Data: Theory and Implications. *Psychological Review*, 123(2), 208-218.

Reviewer #2 (Remarks to the Author):

This manuscript explores the effect of decision processes on the estimation of kernels with psychophysical reverse correlation. Psychophysical reverse correlation is a powerful technique that is becoming widely used, so furthering our understanding is a valuable contribution. The most important contribution is a thorough exploration of how the decision-making process in reaction time tasks influences kernel recovery. This has been carefully done and thoughtfully presented. My most significant concern relates to the treatment of fixed duration tasks. I believe this is only a semantic disagreement, so it should be simple to resolve. Nonetheless, as currently written, I think the discussion of fixed duration tasks is misleading and unhelpful. They are treated as problematic because the kernels do not reveal the "true sensory weights". Obviously, this depends on how one defines "true sensory weight" – something the authors conspicuously fail to do. If one were to define this such that any temporal dynamics in the kernel reflect the temporal dynamics of sensory filters, one could defend their subsequent statements. But that is a straw man that misrepresents the existing literature. It has long been recognized that decision processes, such as reaching a bound, can influence the dynamics of recovered kernels (and importantly, not their shape in other stimulus dimensions). The simulation that generated figure 3g illustrates why calling this a change in "sensory weight" (or not) is semantically unclear. Here, the decision bound is reached before the end of some trials, and the model subject stops integrating sensory information. For the rest of those trials, the weight connecting sensory input to the decision is set to zero. So, the resulting kernel correctly reflects the mean sensory weight between the stimulus and the decision. But the authors use a definition of "sensory weight" in which it is unchanged after the decision bound. To describe a sensory weight as unchanged when sensory information is suddenly given no weight at least needs careful justification. If they were to say that kernel dynamics do not necessarily reflect the temporal kernel of early sensory neurons that would be a much clearer statement of what I think they mean. But this definition makes it harder to suggest (as they do on pp. 16-17) that previous authors were confused here. Previous authors may have used different language, but in most cases the conclusions drawn from kernel dynamics did not depend on assuming they were measures of sensory temporal filters.

A simple thought experiment clarifies how treacherous the terminology is here. Supposing subjects closed their eyes when the decision bound was reached. This is indistinguishable from stopping evidence accumulation. But surely no-one would argue that the sensory weight changes when the eyes close.

As a result, I strongly disagree with their claim that "Because of variability of decision times, experiments in which stimulus viewing duration is controlled by the experimenter tend to be unsuitable for reliable estimation of the temporal dynamics of sensory weights" (p. 3, line 22). Under an appropriate definition of "sensory weight" (the effective weight of stimulus features the decision) I would argue the opposite is true - controlled durations deliver this measure without all

the complications that arise in RT tasks. The measure does not simply reflect early sensory temporal filters for sure, but that's exactly why terms like "sensory weight" have been used. Of course, I agree that applying their more sophisticated model to RT data provides a richer and powerful description, and may allow stronger statements about sensory filters. But this does not mean that fixed duration kernels are unsuitable.

A closely related point, is that all of their analyses show that they can estimate properties of the decision process using kernels in reaction time. They do NOT show that when this is combined with temporal dynamics in sensory filters that they can separate the two. Since the putative limitation of fixed duration experiments that they are "unsuitable for reliable estimation of the temporal dynamics of sensory weights", they need to show that they can do this step in RT tasks.

That's a long discussion for a semantic point, for which I apologize. But it shows how very important it is that the meaning of "true sensory weight" be clearly spelled out, and then used consistently.

Other points.

In the analysis of RT tasks, there are certain places where some useful simplifications could be pointed out. The situation is very different for fixed vs variable NDTs. In figure 3, d-h are for zero NDT, and i-l are for a non-zero AND variable NDT. With a non-zero but fixed NDT, two simple things could be shown. First, referenced to response time, there will be a sudden drop at 0. Second, referenced to stimulus onset there will be a linear decline over the period of the NDT. And a kernel built with this period removed from all trials would be undistorted. So it's really variability in the NDT that is problematic. Finally, in figure 3k, plotting the distribution of NDTs would allow readers to appreciate that the falling tail on the right simply reflects the cumulative distribution of NDTs.

Since variability in the NDT is such a crucial parameter for the kernel modelling, I would like to see a clearer discussion of how well other aspects of the behavioral data constrain this estimate. My understanding of the models fit in figs 4 and 5 is that the model was fit to all of the data. I am not clear that the full distribution of the RTs was even used to constrain estimates of NDT variability. If it's not, then the kernel fits and the chronometric/neurometric fits are really constrained by separate aspects of the data. That means that they can fit the kernel dynamics by invoking NDT variability, but nothing in the data provides an independent confirmation that this is the correct account.

Smaller points.

P 12 line 33 "Neurons representing different choices tend to be negatively correlated". This is NOT typically true of sensory neurons contributing to different choices (e.g. in MT).

We thank the reviewers for their insightful and helpful comments. We have taken them to heart and revised the manuscript accordingly. The suggested changes have significantly improved the paper. We are grateful.

Reviewer #1 (Remarks to the Author):

Psychophysical reverse correlation is a commonly used method to infer the sensory weighting of stimulus information to inform a perceptual decision. Under assumptions of signal detection theory, reverse correlation has been shown to accurately recover true sensory weighting.

However, in this interesting and important article, the authors provide a strong case that reverse correlation does not provide an accurate recovery of true sensory weighting under a variety of common assumptions about decision-making in the sequential sampling framework.

Using a set of stimulations combined with empirical data, the authors show that various characteristics of decision-making distort the reverse correlation recovery of sensory weighting. These characteristics include, most importantly, non-decision time and non-decision time variability, between- and within-trial adjustment of response caution, and to a lesser extent also information leakage, mutual inhibition of accumulators, a reflective bound, or a non-perfect anti-correlation between input of accumulations.

The authors conclude that reverse correlation thus not only reflects sensory weighting, but also characteristics of decision-making. These distortions can be sufficiently strong to warrant serious caution in the interpretation of psychophysical kernels. In light of these findings, the authors illustrate how some previous findings using reverse correlations may need reinterpretation.

At the same time, the authors argue that various characteristics of decision-making aspects have key signatures in the psychophysical reverse correlation. Proper partitioning of the contribution of sensory and decision-making influences on the reverse correlation can still allow meaningful interpretation of reverse correlations.

Taken together, the authors adequately address an interesting research question. However, I have some remaining questions and comments. I am confident that the authors can address these and recommend publication pending major revision.

We thank the reviewer for careful assessment of our paper and wise comments. S/he raises key points that we address in the revised manuscript, as explained below.

Major

1. In the Discussion (p. 16 L. 44 and further), the authors suggest that it is possible to disentangle the influences of sensory and decision-making characteristics, allowing for meaningful interpretation of psychophysical kernels. The reasoning seems to be that researchers could fit a decision-making model to reaction time and accuracy data (e.g., the DDM), use the fitted model to predict a psychophysical kernel, and compare the predicted kernel with the empirically obtained kernel. The key signatures, provided in the Results, could then be used to hypothesize which additional (decision-making) mechanisms took place.

However, I am not convinced of this approach for two reasons. First, the reasoning does not seem to allow the researcher to reject the assumption that sensory weighting is static over time: static sensory weighting is assumed by the DDM that is fitted, and any deviations between the predicted and empirically obtained kernels is interpreted in terms of decision-making effects other than sensory weighting (i.e., in terms of the effects of the decision-making characteristics covered in the Results / Figure 6). Could the use of reverse correlation then also falsify the assumption of static sensory weighting?

That is an excellent question and we believe the answer is yes. In retrospect, we see how our choice of example models with static weights in the previous version of the manuscript could have created the impression that any deviation between the predicted and empirically obtained kernels must be due to the decision-making process. In the revised manuscript, we correct this issue and take two steps to address the reviewer's question. First, we simulate conditions in which sensory weights change dynamically throughout the trial and show our modeling approach can recover the weight dynamics (new Fig. S5). Second, we implement a series of nested models that incorporate time-varying weights for the direction discrimination and face discrimination tasks, showing that the majority of subjects have minimal or no change in their sensory weights within a trial. Because the same models could accurately recover weight dynamics in the simulated data, we do not think our observation about static weights in the experimental data is caused by a low power for the detection of weight dynamics or a fundamental bias in the model to attribute changes of psychophysical kernels to the decision-making process. The key insight from these analyses is that the distribution of choice and reaction times, together with the shape of psychophysical kernels provide adequate constraints for inferring both the decision-making mechanism and temporal dynamics of sensory weights. These points are now explained in the manuscript (pages 11-12, newly added section: "Testing for temporal dynamics of sensory weights", and Fig. S5).

Second, this approach does not seem to consider the possibility of a mixture of decision-making mechanisms influencing the reverse correlation at the same time. The key signatures provided in Figure 6 are obtained when considering a single decision-making effect in isolation of all others. It remains an open question whether these 'key signatures' are still recognizable if multiple decision-making characteristics operate at the same time (e.g., non-decision time variability combined with a change in threshold). Illustrating this complexity, the authors show on P.14-15 and Figure 7 that combining multiple characteristics can lead to complex distortions in the psychophysical reverse correlation, even including the possibility that multiple decision-making effects balance each other out. Given the variability of effects and their similarity, are the effects of decision-making mechanisms on the reverse correlation identified?

This is an important concern. The answer to the reviewer's question is a qualified yes. It is true that multiple factors can have similar effects on the shape of psychophysical kernels, and that factors with opposing effects can cancel each other out, making it hard to infer the underlying mechanisms solely based on the dynamics of psychophysical kernels. However, the approach suggested in the manuscript goes a long way to tease apart different mechanisms. Specifically, the shape of the kernel should be always interpreted in conjunction with the RT and choice distributions. Factors that have a similar effect on the kernel often have contrasting effects on choices and RTs. For example, variability of non-decision time, input correlation, and mutual inhibition can all lead to stimulus-aligned kernels that shrink with time. However, as we show in the new Fig. S9, these three mechanisms (i) have different effects on the shape of the tail of the RT distribution, (ii) make different predictions about accuracy for the same sensitivity function (new Fig. S9a), and (iii) differ in the exact shape of the psychophysical kernels (also note the small but detectable, qualitative and quantitative differences in the dynamics of the kernels in Fig. 6). The combination of these signatures enables separation of different mechanisms. A similar contrast in kernel dynamics and RT and choice distributions exists for dropping bounds and reflective bounds, which both cause an inflation of psychophysical kernels over time. Along the same lines, when combinations of different mechanisms with opposing effects lead to a flat kernel (e.g., Fig. 7d), the distribution of RT and choice is often different compared to when the kernel is flat because none of those mechanisms contribute to the decision-making process or a different combination of parameters flattens the kernel (new Fig. S9b). Overall, a three-pronged approach based on the shape of the kernel, and RT and choice distributions makes a powerful

technique for uncovering the mechanisms that shape behavior. However, there are also important limitations on what and how much can be learned just from behavior. For example, when tails of RT distribution play a key role in distinguishing different mechanisms, the amount of available data and presence or absence of experimental manipulations that may encourage fast responses will influence our ability to arrive at the correct conclusions; hence, the answer to the reviewer's question is a qualified yes. We now clarify these points in the paper (page 18).

Further adding to this point is that the data presented in Figure 6 are simulated with a high number of trials (106). While this provides a clean/noise-free description of the key signatures, empirical data is noisy (e.g., Figure 4e). Therefore, it is not entirely clear how well the key signatures could be recovered in empirical data, even if they are to occur in isolation of any other influence on the kernels.

We fully agree with the reviewer that limitation in available data can significantly constrain our ability to distinguish different models and mechanisms. However, based on our experience and results presented in the paper, a few hundreds to thousands of trials are often adequate to capture major trends and most influential factors. The very large trial counts in our simulations are only to create idealized and nice figures for readers. A limited number of trials often causes noisy high-frequency fluctuations in the kernels, which can be smoothed out under reasonable assumptions about the underlying processes. Still, we agree that the strength of our proposal depends on the available data, and that a very limited dataset with a small number of trials (tens to a couple hundreds) can significantly hamper our ability to draw meaningful conclusions, as it would for any other model-based approach. We now explain these points in the paper (page 18).

I think the article would improve from further clarification of this approach in the Discussion, or some evidence that this approach is feasible. Or otherwise, I wonder if the authors see any other methods that may be more suitable to explore the sensory weighting of stimuli to inform decisions?

We now discuss the points above in the manuscript (pages 11-12 and 18, and new Figs. S5 and S9) to provide insight about both the strengths and the weaknesses of our proposal. Many thanks to the reviewer for excellent suggestions and nudging us in the right direction.

Minor

1. P. 8, Figure 4b and P. 10, Figure 5d: In both plots, there seems to be a data point missing for stimulus/motion strength 0?

Because subjects received random feedback for the ambiguous stimuli (stimulus strength 0), $P(\text{correct})$ is 0.5 by design. We now explain this point in the caption of Fig. 4b.

2. P. 4, L. 16-17: "One of the simplest and most commonly used decision-making models that takes these intricacies into account (...) is the drift diffusion model". The phrasing here seems to suggest that the (standard) DDM takes all of the 'intricacies' mentioned on lines 11-14 on the same page into account, but this is not the case for urgency. Maybe a slightly different phrasing can prevent confusion for readers not familiar with the DDM.

Thanks. We have rephrased the sentence (page 4).

3. The simulations assume normally distributed non-decision times, and some variations. Usually, distributions of non-decision times are chosen based on mathematical simplicity rather than cognitive theory (e.g., the uniform distributions in the DDM, Ratcliff & Tuerlinckx (2002), and

LBA, Brown & Heathcote (2008)). However, recent advances have made it possible to estimate the shape of the non-decision time distribution (Verdonck & Tuerlinckx, 2016), showing various shapes of non-decision time distributions. I feel this result is very interesting in light of the presented article and could provide the reader further intuitions on the interpretation of Figure S3 specifically.

Excellent point. We now cite Verdonck & Tuerlinckx (2016) where we refer to Fig. S3 and discuss it briefly (page 7).

4. P. 12 L2-3: “This urgency ... over time”, and p.20, L22-23 “For example, ... decision bound”. This mathematical equivalence depends on the exact dynamics of the decision bound (e.g., Hawkins et al., 2015).

Point taken. We agree with the reviewer and now cite Hawkins et al. (2015). There is one issue that may require clarification. The urgency signal that we define in the current manuscript, and in our previous papers (e.g., Churchland et al 2008; Hanks et al 2014) is identical to the collapsing bound model in Hawkins et al (2015). Our definition of urgency signal is different from another definition by Cisek (e.g., Cisek et al 2009; Thura et al 2012) and Ditterich (2006), which is based on a gain parameter for the evidence accumulation process. When the decision is based on a competition between two evidence accumulation processes with anti-correlated inputs and urgency is defined as an additive term in both accumulation processes, one can create mathematically equivalent DDMs with collapsing bounds. These points are now clarified in the paper (pages 12 and 22).

5. P. 12, L.43: “Another commonly ... can go”. As far as I’m aware (and the authors only cite) the Leaky, Competing Accumulator model as the only model that makes this assumption.

More papers have used this variation, for example Kiani et al (Neuron 2014), Zylberberg & Shadlen (bioRxiv 2016), Usher & McClelland (Psych Rev 2001), Teodorescu & Usher (Psych Rev 2013), and biophysically realistic implementations of the evidence accumulation processes (e.g., Beck et al, Neuron 2008; Wang, Neuron 2002). These citations are now added to the manuscript (page 13).

6. The key signatures provided in Figure 6 (and supplementary figure S5 and S6) give an overview of how various decision-making mechanisms/characteristics can distort reverse correlations. The mechanisms under study seem mainly chosen for their neural plausibility. Other decision-making characteristics that are often studied in behavioral experiments are neglected: bias/start point effects, and between-trial variability in start point and drift rate. Since the characteristics in the Results provide a sufficiently convincing case that distortions in reverse correlation can occur, I do not feel that all the effects of bias and bias/drift between-trial variability should be addressed in detail with simulations (which would provide a substantial amount of extra work). However, I wonder if the authors can shed some light in general terms of what could be expected from bias and between-trial variabilities? Or is there any reason to believe these characteristics may not be as important as the effects considered? (Since the proportionality constant is only valid assuming equal bounds (derivation Eqs. 3-11), all threshold effects on the reverse correlation would presumably be different when a bias is present?)

That is a great point. Bias can change the shape of the kernels. Implementation of the bias by a change in the starting point of the accumulation process can inflate the initial part of the stimulus-aligned kernels. Implementation of the bias by a change in drift rate or by a dynamic bias signal (e.g., Hanks et al 2011) can cause a DC offset in the psychophysical kernel. Trial-to-trial

variability of starting points does not systematically distort the psychophysical kernel. We now show these results in Fig. S8 and discuss them in page 15.

7. It seems that Figure 6j-l are special cases of Figure 6m-o, where mutual inhibition > leakage. In the two-accumulator version, the leaky competing accumulator model (Usher & McClelland, 2001, their Eqs. 5-6) reduces to an OU-process, where the difference between mutual inhibition and leakage determines the dynamics of the evidence accumulation process. Wouldn't it be simpler to illustrate both the effects of mutual inhibition and leakage with Figure 6m-o, and remove Figure 6j-l? Or does Figure 6j-l provide details not present in Figure 6m-o?

Thank you for the good suggestion. We have removed Fig. 6j-l to occupy less real state for the figures.

References

Brown, S.D., & Heathcote, A. (2008) The simplest complete model of choice reaction time: Linear ballistic accumulation. *Cognitive Psychology*, 57, 153-178.

Hawkins, G. E., Forstmann, B. U., Wagenmakers, E.-J., Ratcliff, R., & Brown, S. D. (2015). Revisiting the Evidence of Collapsing Boundaries and Urgency Signals in Perceptual Decision-Making. *The Journal of Neuroscience*, 35, 2476-2484.

Ratcliff, R., & Tuerlinckx, F. (2002). Estimating parameters of the diffusion model: Approaches to dealing with contaminant reaction times and parameter variability. *Psychonomic Bulletin & Review*, 9(3), 438-481.

Usher, M. & McClelland, J. L. (2001). The time course of perceptual choice: The leaky, competing accumulator model. *Psychological Review*, 108(3), 550-592.

Verdonck, S., & Tuerlinckx, F. (2016), Factoring Out Nondecision Time in Choice Reaction Time Data: Theory and Implications. *Psychological Review*, 123(2), 208-218.

Reviewer #2 (Remarks to the Author):

This manuscript explores the effect of decision processes on the estimation of kernels with psychophysical reverse correlation. Psychophysical reverse correlation is a powerful technique that is becoming widely used, so furthering our understanding is a valuable contribution. The most important contribution is a thorough exploration of how the decision-making process in reaction time tasks influences kernel recovery. This has been carefully done and thoughtfully presented.

Thank you for careful assessment of our paper and insightful comments.

My most significant concern relates to the treatment of fixed duration tasks. I believe this is only a semantic disagreement, so it should be simple to resolve. Nonetheless, as currently written, I think the discussion of fixed duration tasks is misleading and unhelpful. They are treated as problematic because the kernels do not reveal the “true sensory weights”. Obviously, this depends on how one defines “true sensory weight” – something the authors conspicuously fail to do. If one were to define this such that any temporal dynamics in the kernel reflect the temporal dynamics of sensory filters, one could defend their subsequent statements. But that is a straw man that misrepresents the existing literature. It has long been recognized that decision processes, such

as reaching a bound, can influence the dynamics of recovered kernels (and importantly, not their shape in other stimulus dimensions). The simulation that generated figure 3g illustrates why calling this a change in “sensory weight” (or not) is semantically unclear. Here, the decision bound is reached before the end of some trials, and the model subject stops integrating sensory information. For the rest of those trials, the weight connecting sensory input to the decision is set to zero. So, the resulting kernel correctly reflects the mean sensory weight between the stimulus and the decision. But the authors use a definition of “sensory weight” in which it is unchanged after the decision bound. To describe a sensory weight as unchanged when sensory information is suddenly given no weight at least needs careful justification. If they were to say that kernel dynamics do not necessarily reflect the temporal kernel of early sensory neurons that would be a much clearer statement of what I think they mean. But this definition makes it harder to suggest (as they do on pp. 16-17) that previous authors were confused here. Previous authors may have used different language, but in most cases the conclusions drawn from kernel dynamics did not depend on assuming they were measures of sensory temporal filters.

A simple thought experiment clarifies how treacherous the terminology is here. Supposing subjects closed their eyes when the decision bound was reached. This is indistinguishable from stopping evidence accumulation. But surely no-one would argue that the sensory weight changes when the eyes close.

As a result, I strongly disagree with their claim that “Because of variability of decision times, experiments in which stimulus viewing duration is controlled by the experimenter tend to be unsuitable for reliable estimation of the temporal dynamics of sensory weights” (p. 3, line 22). Under an appropriate definition of “sensory weight” (the effective weight of stimulus features the decision) I would argue the opposite is true - controlled durations deliver this measure without all the complications that arise in RT tasks. The measure does not simply reflect early sensory temporal filters for sure, but that’s exactly why terms like “sensory weight” have been used. Of course, I agree that applying their more sophisticated model to RT data provides a richer and powerful description, and may allow stronger statements about sensory filters. But this does not mean that fixed duration kernels are unsuitable.

Thanks for raising this key issue, which we think originated from our failure to communicate two key concepts in the previous version of the manuscript. Both are corrected in this revision. First, we agree with the reviewer that psychophysical reverse correlation approximates the effective association of the stimulus and choice, and that such descriptive explanations are valid regardless of the mechanisms that underlie the choice. Consequently, we are not questioning thoughtful past experiments or careful use of psychophysical reverse correlation as a description of the effect of stimulus on choice. Rather, our goal is to push deeper and ask whether psychophysical reverse correlation can go beyond such descriptions and provide insight about the mechanisms that underlie behavior. We try to elevate psychophysical reverse correlation from a technique that reveals only effective associations of stimulus and choice to a technique that reveals the inner working of the sensory and decision-making processes that underlie the choice. We show that it is possible to do so and provide a recipe to achieve this goal. Of course, there are limitations too. However, we show that under reasonable assumptions supported by past studies, one can go quite far to tease apart different sensory and decision-making mechanisms. The point about the validity of psychophysical kernels as a description of stimulus-choice relationship is now explained in Results (page 5) and Discussion (page 19).

The second point that we failed to explain in the previous version but clarify in the current one is the contrast between fixed duration and RT tasks. As mentioned above, we do embrace the reviewer’s perspective and had no intention to dismiss careful use of psychophysical reverse correlation in past studies. Indeed, past studies that used reverse correlation with a fixed duration experimental design significantly advanced the field. The point that we meant to make was that if

one's goal is to distinguish sensory and decision-making mechanisms that shape psychophysical kernels, fixed duration designs have limited utility because they restrict experimenters' ability to determine the beginning and end of the decision-making process. There are remedies for these limitations and also there are successful examples of how these remedies have been used in the past (we cite some). However, RT tasks provide an easier path for mechanistic studies of psychophysical kernels because they enable experimenters to measure reaction times. We now explain this point in Introduction (page 3) and Results (page 5).

Finally, we follow the reviewer's recommendation and define "true sensory weights" at the beginning of Results (page 5). The modeling framework and examples in the paper provide further clarification.

A closely related point, is that all of their analyses show that they can estimate properties of the decision process using kernels in reaction time. They do NOT show that when this is combined with temporal dynamics in sensory filters that they can separate the two. Since the putative limitation of fixed duration experiments that they are "unsuitable for reliable estimation of the temporal dynamics of sensory weights", they need to show that they can do this step in RT tasks.

That is a great question, which is also raised by reviewer #1 (please see our response to R1's first major point). We now modify our fits to experimental data to create a series of nested models that test for a combination of dynamic sensory weights and decision-making processes (Methods, pages 30 and 31). Further, we have added a new figure to demonstrate that the effects of dynamic sensory weights and decision-making processes can be segregated by a model-based approach (new Fig. S5). This segregation is possible because changes in sensory weights and various factors of the decision-making process make contrasting predictions about the distribution of RTs and choices, and details of kernel dynamics. A three-pronged approach based on RTs, choices, and kernels can separate many possible combinations of sensory weights and decision-making processes (new Fig. S9). However, there are important limitations too. These points are expanded and discussed in pages 11-12 (new section: "Testing for temporal dynamics of sensory weights") and page 18.

That's a long discussion for a semantic point, for which I apologize. But it shows how very important it is that the meaning of "true sensory weight" be clearly spelled out, and then used consistently.

Many thanks for the insight and for helping us improve the text and clarify the main concepts. As mentioned above, we explicitly define "true sensory weights" in page 5.

Other points.

In the analysis of RT tasks, there are certain places where some useful simplifications could be pointed out. The situation is very different for fixed vs variable NDTs. In figure 3, d-h are for zero NDT, and i-l are for a non-zero AND variable NDT. With a non-zero but fixed NDT, two simple things could be shown. First, referenced to response time, there will be a sudden drop at 0. Second, referenced to stimulus onset there will be a linear decline over the period of the NDT. And a kernel built with this period removed from all trials would be undistorted. So its really variability in the NDT that is problematic. Finally, in figure 3k, plotting the distribution of NDTs would allow readers to appreciate that the falling tail on the right simply reflects the cumulative distribution of NDTs.

That is a great observation. We have clarified this point in the text (page 7), emphasizing that the *variability* of non-decision time is the key factor to keep in mind when interpreting psychophysical kernels. We have added a new row to Fig. S3 to illustrate this point and explain it in the main text (page 7).

Since variability in the NDT is such a crucial parameter for the kernel modelling, I would like to see a clearer discussion of how well other aspects of the behavioral data constrain this estimate. My understanding of the models fit in figs 4 and 5 is that the model was fit to all of the data. I am not clear that the full distribution of the RTs was even used to constrain estimates of NDT variability. If its not, then then the kernel fits and the chronometric/neurometric fits are really constrained by separate aspects of the data. That means that they can fit the kernel dynamics by invoking NDT variability, but nothing in the data provides an independent confirmation that this is the correct account.

All model fits were done using the RT and choice distributions. Consequently, estimation of the variability of non-decisions times was determined by the trial-to-trial variability of RTs. We now clarify this point in Methods (page 30) and reiterate it in Results (page 11) and Discussion (page 18).

Smaller points.

P 12 line 33 “Neurons representing different choices tend to be negatively correlated”. This is NOT typically true of sensory neurons contributing to different choices (e.g. in MT).

The reviewer is correct about positive “noise” correlations in MT. Similarly positive noise correlations are observed in frontoparietal neurons that represent accumulation of evidence (e.g., Kiani et al, Neuron 2015). However, the referenced sentence is about the correlation of the overall responses across neurons (signal + noise). Responses of MT neurons that represent opposite motion directions or frontoparietal neurons that represent opposite saccades tend to be negatively correlated once stimulus/task induced responses (signal) are taken into account. One group of neurons increases their responses, while the other group decreases their responses. However, the two groups are not perfectly anti-correlated. One of the reasons is the positive noise correlation, as mentioned by the reviewer. We now clarify this point in the text (page 13).

REVIEWERS' COMMENTS:

Reviewer #1 (Remarks to the Author):

The authors have addressed all of my remaining comments and I advise to accept the manuscript as is.

Reviewer #2 (Remarks to the Author):

This paper has been significantly improved, and I have no further concerns.